# VISION-CENTRIC ACTIVATION AND COORDINATION FOR MULTIMODAL LARGE LANGUAGE MODELS

## ABSTRACT

Multimodal large language models (MLLMs) integrate image features from visual encoders with LLMs, demonstrating advanced comprehension capabilities. However, mainstream MLLMs are solely supervised by the next-token prediction of textual tokens, neglecting critical vision-centric information essential for analytical abilities. To track this dilemma, we introduce **VaCo**, which optimizes MLLM representations through **V**ision-Centric **a**ctivation and **Co**ordination from multiple vision foundation models (VFMs). VaCo introduces visual discriminative alignment to integrate task-aware perceptual features extracted from VFMs, thereby unifying the optimization of both textual and visual outputs in MLLMs. Specifically, we incorporate the learnable *Modular Task Queries* (MTQs) and *Visual Alignment Layers* (VALs) into MLLMs, activating specific visual signals under the supervision of diverse VFMs. To coordinate representation conflicts across VFMs, the crafted *Token Gateway Mask* (TGM) restricts the information flow among multiple groups of MTQs. Extensive experiments demonstrate that VaCo significantly improves the performance of different MLLMs on various benchmarks, showcasing its superior capabilities in visual comprehension.

## 1 INTRODUCTION

Benefiting from advancements in LLMs OpenAI (2022); Touvron et al. (2023), MLLMs have exhibited remarkable capabilities in various visual understanding tasks, such as visual question answering Achiam et al. (2023); Liu et al. (2023a), object grounding Wang et al. (2023a), and referring expression comprehension You et al. (2023). These innovative models typically incorporate images as multimodal tokens into LLMs, utilizing features from pre-trained vision-language models (*e.g.*, CLIP Radford et al. (2021)) to facilitate language-output understanding. Within this architecture, although the effectiveness of advanced LLMs in enhancing multimodal capabilities is well-established, the potential of visual components remains insufficiently underscored, posing a bottleneck to achieving comprehensive visual understanding Wang et al. (2024a); Tong et al. (2024a).

To advance the integration of vision signals, various approaches Jain et al. (2024); Tong et al. (2024a) strive to effectively capture multimodal conditional and marginal distributions by ensembling multiple vision foundation models (VFMs), seeking improved compatibility with the linguistic priors inherent in LLMs. However, these models present challenges such as *increased computational demands* from the extra visual encoder Tong et al. (2024b); Jain et al. (2024); Lu et al. (2024) and *significant alignment burdens* from complex visual connectors Tong et al. (2024a); Fan et al. (2024). Critically, directly feeding multiple VFM features into MLLMs inherently leads to the degradation of rich visual information due to the lack of visual supervision Wang et al. (2024a); Huang et al. (2025). Therefore, ROSS Wang et al. (2024a) imposes a reconstructive objective for the image tokens output by MLLMs, intrinsically activating vision-centric representations, and eliminating the reliance on external modules. However, the self-supervised scheme focuses on low-level features through the vision token restoration, rather than effectively capturing the diverse perceptual semantics crucial for comprehensive visual understanding. Moreover, vision experts Oquab et al. (2023); Yang et al. (2024a); Wang et al. (2025) tailored for specific perceptual tasks demonstrate the potential to deliver fine-grained priors for visual understanding, whereas the conflicts Metz et al. (2020); Vandenhende et al. (2021) arising from their task-aware representations may hinder their complementary integration. Consequently, the essence of effectively incorporating multiple task-aware perceptual priors into MLLMs lies in: (**i**) preventing the gradual diminishing of perceptual cues directly fed into the MLLM; and (**ii**) avoiding conflicts between multiple task-aware representations.

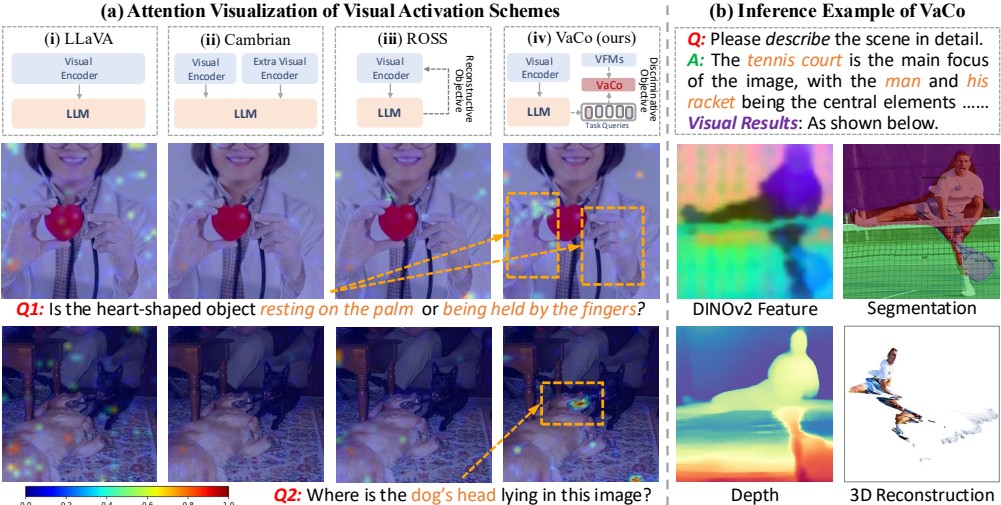

Figure 1: **Visualization of Visual Activation Methods**: (a) **Attention Visualization**. Utilizing samples from MMVP Tong et al. (2024b), we investigate which visual regions dominate the final predictions of MLLMs. Specifically, since MLLMs are anticipated to deliver answers at the final position of the input sequence, we visualize the attention scores between the last token and all image tokens. Compared to the other schemes (*i.e.*, (i) LLaVA Liu et al. (2023a), (iii) multi-encoder scheme Tong et al. (2024a), (iv) reconstructive scheme Wang et al. (2024a) and (iv) the proposed VaCo), our vision-centric activation effectively directs MLLMs to focus on the crucial visual areas mentioned in the questions (*Q*) (*e.g.*, "*palm*" and "*fingers*" in the first example), leading to correct answers; (b) **Inference Example**. VaCo facilitates MLLM in delivering comprehensive answers (*A*) through the assistance of diverse *Visual Results*. Note that visual outputs are an *optional byproduct*.

To address these issues, we propose **VaCo**, a tuning approach for MLLMs that leverages **V**ision-Centric **a**ctivation and **Co**ordination via existing VFMs, facilitating the vision-understanding capabilities of MLLMs, as illustrated in Figure 1 (a). Specifically, we advocate providing supplementary *query-based* visual supervision, emphasizing *discriminative* alignment objectives across various VFMs to preserve perceptual cues, as opposed to the *generative* objective adopted in recent work Wang et al. (2024a). The primary challenge revolves around converting MLLM representations into distinct task-aware spaces, facilitating the activation of comprehensive visual knowledge under the supervision of multiple VFMs. To this end, we design learnable Modular Task Queries (MTQs) (*e.g.*, depth queries) for specific VFM and feed them together with the vision token into the MLLM. These MTQs aim to extract task-specific visual information from the visual representations within MLLMs. Further-

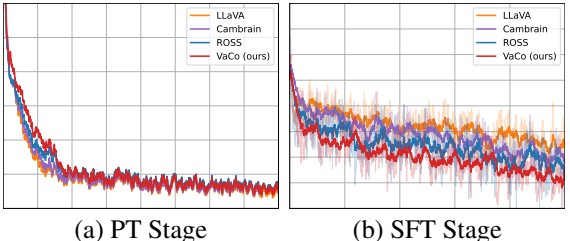

(a) PT Stage      (b) SFT Stage

Figure 2: **Textual Loss Visualization** across two stages. We investigate the impact of visual signal activation on text output: (a) At Pre-Training (PT) stage, the integration of a pure vision model in VaCo complicates initial alignment, but the loss convergence is similar to others; and (b) at Supervised Fine-Tuning (SFT) stage, the textual loss of VaCo converges more rapidly to a lower value, highlighting that query-based visual activation and coordination effectively facilitate MLLM acquisition of comprehensive visual information.

more, we introduce Visual Alignment Layers (VALs), which convert MTQs into Visual Alignment Queries (VAQs) that are dimensionally aligned with a specific vision task. Through the *collaboration of modular task and visual alignment queries*, the causally structured MLLM is instructed to first output MTQs encapsulating specific perceptual priors before generating the final answer, as presented in Figure 1 (b). Thus, the query-based VaCo indirectly activates fine-grained vision information within the original MLLM representations. Although we employ separate queries for different VFMs, representational conflicts Metz et al. (2020); Vandenhende et al. (2021) still poten-

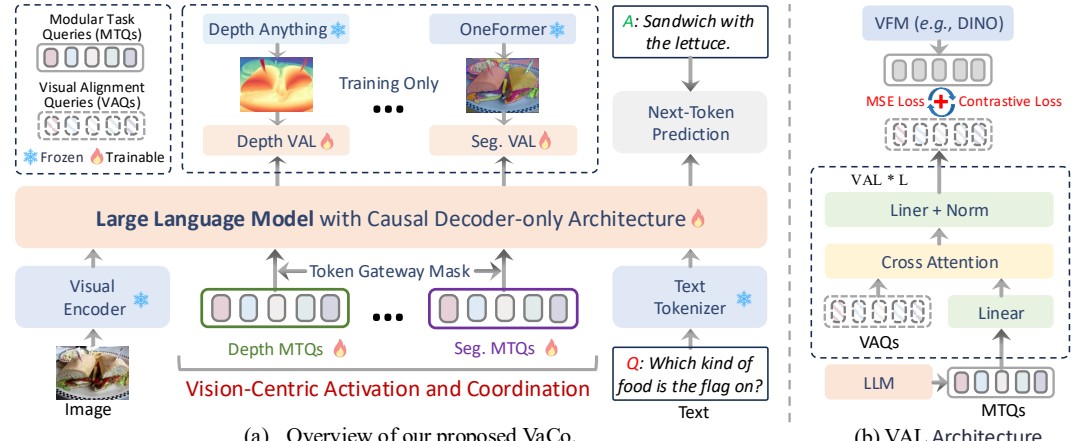

(a) Overview of our proposed VaCo.  (b) VAL Architecture.

Figure 3: **Framework Overview** of our VaCo. (a) We define the Modular Task Queries (MTQs) for various vision tasks, which jointly input the LLM with image and text tokens. The Visual Alignment Layer (VAL) directs these queries towards specific VFMs (e.g., Depth Anything Yang et al. (2024a)). We develop Token Gateway Mask (TGM) for causal attention to prevent feature conflicts across multiple MTQs. (b) Detailed architecture of VAL, with each equipped with an independent Visual Alignment Queries (VAQs), establishes a collaborative mechanism with MTQs.

tially arise among multiple groups of MTQs processed by the causal MLLM. Therefore, we present the Token Gateway Mask (TGM) to restrict the information flow based on MTQ type, coordinating the diverse representations across multiple VFMs. As shown in Figure 2, VaCo promotes the convergence of the next-token prediction for textual tokens, showcasing its effectiveness in activating and coordinating visual priors in the two-stage MLLM training framework Liu et al. (2023a).

In summary, our key contributions are as follows: **(i)** We emphasize the crucial role of visual supervision in MLLMs, suggesting that *query-based* discriminative alignment objectives are more effective than generative ones when integrating multiple VFMs; **(ii)** We present distinct learnable Modular Task Queries (MTQs) and Visual Alignment Layers (VALs) to activate perceptual cues in MLLMs via various VFMs, complemented by the Token Gateway Mask (TGM) to coordinate task-aware representations within the causal architecture. **(iii)** We conduct extensive experiments on diverse visual-language benchmarks, spanning general, region-level, and scene-level tasks, which highlight the superior multimodal understanding of the proposed VaCo.

## 2 PRELIMINARY

**Multimodal Large Language Models**. A LLM Radford et al. (2018; 2019) parameterized by $\theta$ aims to model the canonical causal distribution $p_\theta(\boldsymbol{x})$ over a sequence of text tokens $\{\boldsymbol{x}_i\}_{i=1}^{T}$ with length $T$, denoted as $p_\theta(\boldsymbol{x}) = \prod_{t=1}^{T} p_\theta(\boldsymbol{x} \mid \boldsymbol{x}_{<t})$. To achieve effective multimodal understanding, MLLMs Liu et al. (2023a) expand the original text tokens with $K$-length visual tokens $\{\boldsymbol{v}_i\}_{k=1}^{K}$ as the input sequences of LLM. A typical method to obtain these visual tokens involves processing image $\boldsymbol{I}$ through the patch embedding of a $\zeta$-parameterized visual encoder $\mathcal{E}_\zeta$, such as CLIP Radford et al. (2021) and SigLIP Zhai et al. (2023). Subsequently, a $\phi$-parameterized multimodal projector $\mathcal{P}_\phi$ is employed to map these visual tokens into the feature space of LLMs. Then, the maximum likelihood estimation (MLE) objective function of an MLLM can be formulated as follows:

$$\mathcal{L}_{\text{MLLM}}(\Theta = \{\theta, \phi\}, \boldsymbol{x}, \boldsymbol{I}) = -\mathbb{E}_t[p_\theta(\boldsymbol{x}) = \prod_{t=1}^{T} p_\theta(\boldsymbol{x} \mid \boldsymbol{x}_{<t}, \boldsymbol{v})] \tag{1}$$

where $\Theta = \{\theta, \phi\}$ is the optimized parameters and $\boldsymbol{v} = \mathcal{P}_\phi \circ \mathcal{E}_\zeta(\boldsymbol{I})$ is projected visual tokens. Typically, the optimization of MLLMs involves two stages, *i.e.*, the pre-training of the projector connecting the visual encoder and LLM, and the instruction tuning of both the projector and LLM.

## 3 METHODOLOGY

As depicted in Figure 3, we present **VaCo**, an innovative **V**ision-Centric **a**ctivation and **Co**ordination method for tuning MLLMs. This method seeks to improve the visual-language understanding of MLLMs by harnessing visual priors within VFMs. The VaCo comprises two key components: (1) *Vision-Centric Activation*: We design Modular Task Queries (MTQs) to activate distinct visual priors within MLLMs. This is accomplished by the proposed Visual Alignment Layers (VALs), which convert MTQs into distinct task-aware spaces utilizing discriminative objectives with various VFMs; and (2) *Vision-Centric Coordination*: Token Gateway Mask (TGM) restricts information flow among multiple groups of MTQs, preventing representation conflicts across diverse task-specific VFMs.

### 3.1 VISION-CENTRIC ACTIVATION

**Modular Task Queries for Visual Activation.** When MLLMs seek to interpret a scene according to the given instruction (*e.g.*, *"Which kind of food is the flag on?"*), visual information (*e.g.*, depth and semantics) intuitively plays a crucial role. Intuitively, directly feeding multiple VFM features into MLLMs is expected to enhance scene understanding. However, explicit integration methods can inherently degrade rich visual information due to insufficient visual supervision. This tendency diminishes the role of multiple visual encoders while substantially increasing visual encoding costs. Thus, we advocate for intrinsic visual activation of

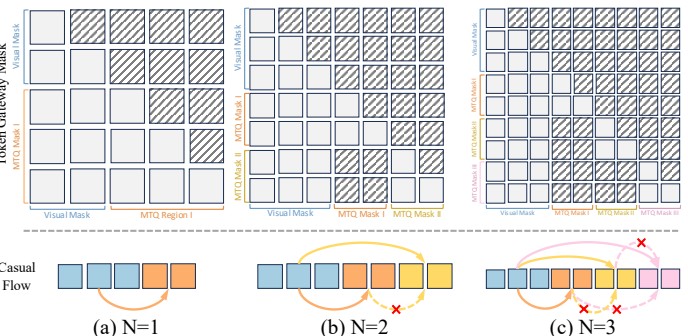

Figure 4: **Illustration of the Token Gateway Mask (TGM)**. Here we present the TGM varying the number of visual tasks, *i.e.*, $N = 1, 2, 3$. **Above** illustrates the construction of TGM, while **below** depicts the information flow within a causal transformer utilizing TGM. We represent the image token in blue, while different types of MTQs are depicted in orange, yellow and pink, respectively. Note that we omit the text tokens for simplicity.

MLLMs under the supervision of diverse VFMs. For efficient integration of multiple vision models, we introduce Modular Task Queries (MTQs) into MLLMs to distill task-specific information. The design of modular queries aim to facilitate understanding capability with multi-source visual supervision, thereby distinguishing from the scheme that directly employs *single* supervision for visual tokens. Formally, we define a set of learnable modular task queries, denoted by $\boldsymbol{q}_{task} = \{q_i\}_{i=1}^{Q}$, each with a length of $Q$. The subscript $task$ denotes the visual task (*e.g.*, depth and segmentation). For simplicity, we apply this notation until a specific model is designated. Then the input sequences of MLLM can be denoted as $\{\boldsymbol{v}, \boldsymbol{q}, \boldsymbol{x}\}$. To direct these queries towards specific visual tasks, we present the visual alignment layer below.

**Visual Alignment Layer.** To integrate the strengths of existing powerful vision foundation models, we propose the Visual Alignment Layer (VAL) to project modular task queries into the feature space of specific vision-centric models, as illustrated in Figure 3 (b). Specifically, the task-specific VAL comprises a set of learnable Visual Alignment Queries (VAQs) $\boldsymbol{q}_{val} \in \mathbb{R}^{M \times D}$ that are randomly initialized, which is consistent with the length $M$ and dimension $D$ of the image token in the specific visual model. Given the modular task queries $\boldsymbol{q}_{task} \in \mathbb{R}^{Q \times D}$ output by the MLLM and the linear layer, the VAL applies several Transformer blocks to convert them into the feature space of a task-aware vision model. The cross-attention mechanism treats VAQ as the query while considering the MTQs as the key and value. This design not only extracts informative visual representations independent of linguistic instructions, but also intrinsically activates visual perception within MLLM representation. This is because an MLLM with the causal attention mechanism, when integrated with MTQs, prioritizes generating task-specific visual features before providing final answers. In practice, once obtaining output queries $\boldsymbol{q}$ of a $\xi$-parameterized VAL that are dimensionally compatible with the particular VFM, we employ mean squared error (MSE) and contrastive objective (*i.e.*,

InfoNCE He et al. (2020)) to achieve alignment between VAQ and VFM features $\mathcal{F}(\boldsymbol{I})$:

$$\mathcal{L}_{\text{task}} = \mathbb{E}_{\boldsymbol{q}}\left[\|\boldsymbol{q} - \mathcal{F}(\boldsymbol{I})\|_2^2\right] - \frac{\lambda}{B}\sum_{i=1}^{B}\left[\log\frac{\exp\left(\text{sim}(\boldsymbol{q}_i, \mathcal{F}(\boldsymbol{I}_i))/\tau\right)}{\sum_{j=1}^{B}\exp\left(\text{sim}(\boldsymbol{q}_i, \mathcal{F}(\boldsymbol{I}_j))/\tau\right)}\right] \quad (2)$$

where $\mathcal{F}$ denotes the pre-trained VFM, which remains frozen during the tuning process, $B$ denotes the number of samples, $\lambda = 0.1$ is the balancing factor, and $\tau$ denotes the learnable scaling factor for the contrastive item. Through the optimization of this objective, coupled with the synergy between MTQs and VAQs, we activate the existing visual representation of MLLM by indirectly incorporating the expert knowledge from purely visual models without text-visual alignment. Compared to multi-encoder schemes that directly input visual modalities into MLLM, our method relies solely on MTQs during inference without additional visual encoders. Therefore, VaCo efficiently utilizes diverse visual cues while significantly reducing computational costs (see Table 8 of **Appendix**).

### 3.2 VISION-CENTRIC COORDINATION

**Token Gateway Mask.** While our VaCo utilizes distinct VAL for various visual tasks, the task-specific MTQs remain actively involved in interacting within the causal attention mechanism of the MLLM. As indicated in Table 4 (b), the diverse objectives of each task lead to representational conflicts among the task queries as processed by the causal MLLM, resulting in a significant performance degradation. For example, the depth estimation task emphasizes global object hierarchies, while the segmentation task focuses on local geometrical shapes. For stable alignment with multiple visual features, building on the causal decoder-only architecture of MLLMs, we design a Token Gateway Mask (TGM) to restrict the information flow among multiple groups of MTQs, as showcased in Figure 4. This technique guarantees that different MTQs remain invisible to each other during causal processing, permitting interaction exclusively with the image tokens. Together with the individual queries of each task, TGM alleviates the potential representation conflicts across different vision tasks. Such a mechanism coordinates multiple VFMs and modularly activates specific visual priors within MLLMs.

**Options of the VFMs**. Benefiting from the abundant community resources, we select several task-aware VFMs, *i.e.*, Depth Anything V2 Yang et al. (2024a), OneFormer Jain et al. (2023), and VGGT Wang et al. (2025), alongside a self-supervised scheme, *i.e.*, DINOv2 Oquab et al. (2023). These choices are made because collectively, these models address core dimensions of visual understanding, *i.e.*, depth, semantics, geometry, and generalization. Nonetheless, VaCo is compatible with an additional visual model of the Vision Transformer (ViT) architecture.

### 3.3 OPTIMIZATION OBJECTIVE

Given the MTQs, textual responses from MLLMs, and vision-centric targets from VFMs, we tune the image projector, MTQs, VAL, and LLM following a two-stage training recipe Liu et al. (2023a), *i.e.*, the pre-training (PT) stage and the supervised fine-tuning (SFT) stage. The training loss of the MLLM equipped with the proposed MTQs, VALs, and TGM can be summarized as follows:

$$\mathcal{L}(\Theta = \{\theta, \phi, \xi\}, \boldsymbol{x}, \boldsymbol{I}, \boldsymbol{q}_{task}) = \mathcal{L}_{\text{MLLM}} + \sum_{i=1}^{N}\alpha_i\mathcal{L}_{\text{task}}^i, \quad (3)$$

where $N$ is the number of the vision-centric tasks. $\alpha_i$ is a hyperparameter, which is typically set to 1.0. Building on this objective, we train the projector, MTQs, and VALs during the PT stage, and subsequently, while in the SFT stage, we jointly optimize these components along with the LLM.

### 4 EXPERIMENTS

**Implementation Details**. We conduct all experiments based on the LLaVA architecture. In particular, we utilize Vicuna-1.5-7B Chiang et al. (2023) and Qwen2-7B Bai et al. (2023) as backbones, alongside the CLIP-ViT-L/14@336 Radford et al. (2021) and SigLIP-ViT-SO400M/14@384 Zhai et al. (2023) as visual encoders, to assess the effectiveness of our proposed approach. In the VaCo framework, we employ DINOv2 Oquab et al. (2023), OneFormer Jain et al. (2023), Depth Anything V2 Yang et al. (2024a), and VGGT Wang et al. (2025) as the VFMs. For the pre-training and supervised fine-tuning phases, we employ the LLavA-558K Liu et al. (2023a) and Cambrian-737K Tong et al. (2024a) datasets, respectively. To demonstrate the improvement of understanding ability at the region/scene-level, we integrate the AS-V2 datasets Wang et al. (2024b), which annotates the

Table 1: **Performance Comparison** on general visual-language benchmarks. We assessed our model on the following benchmarks, with some names abbreviated: POPE Li et al. (2023c); Hallu.: HaluBench Guan et al. (2024); MMB$^{EN/CN}$: MMBench-English/Chinese Liu et al. (2024b); SEED$^I$: SEED-Image Li et al. (2023a); MMMU Yue et al. (2024); MMVP Tong et al. (2024b); Real.: Real-WorldQA x.ai (2024); and AI2D Hiippala et al. (2021). The best performances are marked in **bold**.

| Method | POPE | Hallu. | MMB$^{EN}$ | MMB$^{CN}$ | SEED$^I$ | MMMU | MMVP | Real. | AI2D |
|---|---|---|---|---|---|---|---|---|---|
| GPT-4V-1106 OpenAI (2023) | 75.4 | 65.8 | 75.8 | 75.1 | 71.6 | 53.8 | 50.0 | 63.0 | 78.2 |
| Gemini-1.5 Pro Team et al. (2023) | - | - | 73.6 | - | 70.7 | 47.9 | - | - | - |
| MM-1-8B McKinzie et al. (2024) | 86.6 | - | 72.3 | - | 69.9 | 37.0 | - | 72.6 | - |
| Mini-Gemini-8B Li et al. (2024b) | - | - | 72.7 | - | 73.2 | 37.3 | 18.7 | 64.5 | 73.5 |
| DeepSeek-VL-7B Lu et al. (2024) | 85.8 | 44.1 | 73.2 | 72.8 | 70.4 | 36.6 | - | - | 64.9 |
| Cambrian-1-8B Tong et al. (2024a) | 87.4 | 48.7 | 75.9 | 68.9 | 74.7 | 42.7 | 51.3 | 64.6 | 73.0 |
| MiniCPM-Llama3-V 2.5 Yao et al. (2024) | - | 42.4 | 77.2 | 74.2 | 72.3 | 45.8 | - | 63.5 | 78.4 |
| *Backbone: Qwen2-7B + Vision Encoder: SigLIP-ViT-SO400M/14@384* | | | | | | | | | |
| LLaVA-1.5-7B Liu et al. (2023a) | 88.5 | 57.3 | 76.3 | 75.7 | 72.3 | 41.8 | 40.7 | 57.9 | 74.0 |
| ROSS-7B Wang et al. (2024a) | **88.7** | **58.2** | 76.9 | 76.3 | 72.1 | 43.8 | 49.3 | 59.1 | 74.5 |
| VaCo-7B (**ours**) | 88.4 | 57.9 | **78.6** | 76.9 | **73.0** | **46.7** | **52.7** | **63.7** | **78.9** |
| *Backbone: Vicuna-1.5-7B + Vision Encoder: CLIP-ViT-L/14@336* | | | | | | | | | |
| LLaVA-1.5-7B Liu et al. (2023a) | 86.2 | 47.5 | 65.5 | 58.5 | 66.0 | 34.5 | 20.0 | 52.7 | 55.4 |
| LLaVA-1.6-7B Liu et al. (2024a) | 86.5 | 35.8 | 67.4 | 60.1 | 70.2 | 35.8 | 37.3 | 54.2 | 67.1 |
| ROSS-7B Wang et al. (2024a) | **87.2** | 55.8 | 67.6 | 59.8 | 66.4 | 34.0 | 36.0 | 53.2 | 61.4 |
| VaCo-7B (**ours**) | 87.0 | **57.2** | **68.5** | **64.2** | **67.1** | **36.8** | **37.9** | **57.6** | **68.3** |
| *Backbone: Vicuna-1.5-13B + Vision Encoder: CLIP-ViT-L/14@336* | | | | | | | | | |
| LLaVA-1.5-13B Liu et al. (2023a) | 82.5 | 44.9 | 68.8 | 63.6 | 68.2 | 36.6 | 32.0 | 63.3 | 60.8 |
| LLaVA-1.6-13B Liu et al. (2024a) | 86.2 | 36.7 | 70.0 | 64.1 | 71.9 | 36.2 | 35.3 | 65.4 | 72.4 |
| Mini-Gemini-13B Li et al. (2024b) | - | - | 68.6 | - | 73.2 | 37.3 | 19.3 | 63.7 | 70.1 |
| Cambrian-1-13B Tong et al. (2024a) | 85.7 | 54.0 | **75.7** | 65.9 | 74.4 | 40.0 | 41.3 | 64.3 | 73.6 |
| ROSS-13B Liu et al. (2023a) | 88.7 | 56.4 | 73.6 | 67.4 | 71.1 | 41.3 | 44.7 | 65.2 | 73.8 |
| VaCo-13B (**ours**) | 88.3 | **57.8** | 75.4 | **68.3** | **74.5** | **43.0** | **46.0** | **68.0** | **75.1** |

question-answering pairs in the formats of grounding and relation. We perform the pre-training phase for 1 epoch, utilizing a learning rate of $2e^{-3}$ and a batch size of 256. We fine-tune the model for 3 epochs, with a learning rate of $2e^{-5}$ and a batch size of 128. The length $Q$ of MTQs for a single task is set to 8. We optimize both phases with the AdamW optimizer Kingma & Ba (2014). We implement all experiments on 8 NVIDIA H20 GPUs, each with 98 GB of memory. We carry out the majority of metric evaluations using the VLEMEval library Duan et al. (2024), except when certain metrics are not covered. For detailed evaluation procedures, please refer to the **Appendix**.

## 4.1 QUANTITATIVE COMPARISON

**Results on General Benchmarks**. To assess the general capabilities of our VaCo, we conduct a comprehensive comparison with prominent MLLMs, as presented in Table 1. The benchmarks encompass diverse sub-tasks that evaluate various fine-grained capabilities, such as logic/attribute/relation understanding and coarse/fine-grained perception in MMBench Liu et al. (2024b). Benefiting from the stronger visual comprehension ability provided by vision-centric activation and coordination, VaCo achieves superior performance on these benchmarks. For instance, our VaCo attains an overall score of 78.6 on MMBench-English and 46.7 on MMMU Yue et al. (2024), outperforming LLaVA-1.5 Liu et al. (2023a) by 2.3 points and 4.9 points, respectively. Notably, compared to ROSS Wang et al. (2024a) that employs *intrinsic visual activation* with a reconstructive objective, our VaCo demonstrates significant advantages on benchmarks such as SEED Li et al. (2023a), MMVP Tong et al. (2024b), RealWorld, and AI2D Hiippala et al. (2021). This indicates that incorporating a query-based discriminative distillation objective with perceptual priors across *multiple* visual foundation models is more effective. Besides, our VaCo allows MLLMs to utilize a single visual encoder during inference, avoiding additional computational overhead from multiple encoders. Despite this, VaCo achieves superior performance across various metrics compared to Cambrian-1 Tong et al. (2024a), which aggregates multiple visual encoders (CLIP Radford et al. (2021), SigLIP Zhai et al. (2023), DINOv2 Oquab et al. (2023), and ConvNext Liu et al. (2022)). These results highlight the efficacy of vision-centric activation and coordination in improving the general visual-language understanding capabilities of MLLMs.

**Results on Region-level Benchmarks**. To assess the effectiveness of VaCo in promoting regional visual comprehension ability, we conduct an evaluation on two representative regional-level tasks: the Referring Expression Comprehension (REC) Kazemzadeh et al. (2014); Mao et al. (2016) and Visual Commonsense Reasoning (VCR) Zellers et al. (2019) benchmarks. In Table 2 (a), we present

Table 2: **Performance Comparison** on *region-level* benchmarks (*i.e.*, Referring Expression Comprehension (REC) Kazemzadeh et al. (2014); Mao et al. (2016) and Visual Commonsense Reasoning (VCR) Zellers et al. (2019) benchmarks). REC evaluates the ability to locate target objects based on a given description, while VCR assesses the commonsense reasoning ability with region referring. **Q**, **A**, and **R** represent the Question, Answer, and Rationale, respectively. → indicates that the model is required to make specific types of choices based on given conditions.

<table>
<tr><td colspan="4" align="center">(a) Results on REC benchmark.</td><td colspan="4" align="center">(b) Results on VCR benchmark</td></tr>
<tr><td rowspan="2">Model</td><td colspan="3" align="center">RefCOCO</td><td rowspan="2">Model</td><td colspan="3" align="center">Validation Accuary (%)</td></tr>
<tr><td>Val</td><td>Test-A</td><td>Test-B</td><td>Q→A</td><td>QA→R</td><td>Q→AR</td></tr>
<tr><td>OFA-L Wang et al. (2022)</td><td>79.96</td><td>83.67</td><td>76.39</td><td>Unicoder-VL Li et al. (2020)</td><td>72.6</td><td>74.5</td><td>54.5</td></tr>
<tr><td>Shikra-13B Chen et al. (2023b)</td><td>87.83</td><td>91.11</td><td>81.81</td><td>VLBERT Su et al. (2019)</td><td>75.5</td><td>77.9</td><td>58.9</td></tr>
<tr><td>Qwen-VL-7B Bai et al. (2023)</td><td>88.55</td><td>92.27</td><td>84.51</td><td>ERNIE-ViL-L Yu et al. (2021)</td><td>78.5</td><td>83.4</td><td>65.8</td></tr>
<tr><td>Ferret-13B You et al. (2023)</td><td>89.48</td><td>92.41</td><td>84.36</td><td>VILLA Gan et al. (2020)</td><td>78.5</td><td>82.6</td><td>65.2</td></tr>
<tr><td>ASMv2-13B Wang et al. (2024b)</td><td>90.56</td><td>94.24</td><td>86.24</td><td>ASMv2-13B Wang et al. (2024b)</td><td>87.8</td><td>88.8</td><td>78.4</td></tr>
<tr><td>VaCo-13B (ours)</td><td>91.63</td><td>95.59</td><td>87.91</td><td>VaCo-13B (ours)</td><td>89.2</td><td>89.8</td><td>79.8</td></tr>
</table>

Table 3: **Performance Comparison** on *scene-level* benchmarks (*i.e.*, (a) Panoptic Scene Graph (PSG) Yang et al. (2022) and (b) Circular-based Relation Probing Evaluation (CRPE) Wang et al. (2024b) benchmarks). We present the triplet recall and mean recall of PSG to assess the open scene graph generation capabilities, while the number of tuples (#Tuple) in the scene graph serves as a metric for evaluating the redundancy of the predictions. CRPE evaluates the scene understanding ability from four perspectives: existence, subject, predicate, and object.

<table>
<tr><td colspan="4" align="center">(a) Results on PSG benchmark.</td><td colspan="5" align="center">(b) Results on CRPE benchmark</td></tr>
<tr><td>Model</td><td>#Tuples</td><td>Recall</td><td>mRecall</td><td>Model</td><td>Exist.</td><td>Subj.</td><td>Pred.</td><td>Obj.</td></tr>
<tr><td>TextPSG Zhao et al. (2023)</td><td>50</td><td>4.8</td><td>-</td><td>Qwen-VL Bai et al. (2023)</td><td>85.1</td><td>45.7</td><td>38.2</td><td>31.6</td></tr>
<tr><td>TextPSG Yang et al. (2022)</td><td>100</td><td>5.5</td><td>-</td><td>LLaVA-1.5 Liu et al. (2023a)</td><td>88.7</td><td>57.4</td><td>54.2</td><td>55.2</td></tr>
<tr><td>ASMv2-13B Wang et al. (2024b)</td><td>9.2</td><td>14.2</td><td>10.3</td><td>ASMv2-13B Wang et al. (2024b)</td><td>92.1</td><td>69.2</td><td>59.0</td><td>65.3</td></tr>
<tr><td>VaCo-13B (ours)</td><td>8.3</td><td>16.1</td><td>12.5</td><td>VaCo-13B (ours)</td><td>96.1</td><td>76.5</td><td>63.3</td><td>70.3</td></tr>
</table>

the comparative results on the REC benchmark, which assesses the ability of the model to locate target objects based on given descriptions. Compared to leading MLLMs like Qwen-VL Bai et al. (2023), our VaCo demonstrates a marked improvement in accuracy scores. We also evaluate the commonsense abilities on the VCR dataset, as shown in Table 2 (b). The VCR is structured as single-choice questions and incorporates region referencing within both the questions and answers. Remarkably, despite the lack of fine-tuning on the VCR dataset under the single-task setting, our VaCo still delivers competitive performance against the current state-of-the-art models.

**Results on Scene-level Benchmarks**. Expanding upon regional-level understanding, further relationship understanding allows for a more comprehensive assessment of the MLLM visual understanding at the scene level. To this end, we conducted scene-level evaluations with Panoptic Scene Graph (PSG) Yang et al. (2022) and Circular-based Relation Probing Evaluation (CRPE) Wang et al. (2024b) dataset. As illustrated in Table 3 (a), we present the compared results on the PSG benchmark. Following the open-ended scene graph generation approach, TextPSG Zhao et al. (2023), we present both the triplet recall and the mean recall for each predicate category (mRecall). We also report the average number of predicted triplets (#Tuple), where a larger #Tuple typically achieves higher performance but tends to produce more redundant outputs. The results indicate that, despite generating fewer tuples, VaCo maintains a competitive Recall of 16.1 and mRecall of 12.5. Besides, we highlight the superiority of our VaCo in grounding and relationship understanding within the CRPE benchmark, as demonstrated in Table 3 (b). We report four CRPE scores (existence, subject, predicate, and object), which evaluate the capabilities of MLLMs in object recognition and relationship comprehension. Our VaCo exhibits a significant enhancement in object and relationship understanding compared to other models. For example, VaCo achieved an existence score of 96.1 and a predicate score of 63.3, significantly outperforming other state-of-the-art MLLMs. These results illustrate that our model, augmented by the vision-centric activation and coordination, effectively comprehends the relationships between objects within the image.

## 4.2 ABLATION STUDY

In Table 4, we investigate the effects of various visual activation strategies and VFMs. We adopt LLaVA-v1.5 Liu et al. (2023a) baseline for all ablation studies, which integrates Vicuna-7B as the backbone and CLIP-ViT-L/14@336 as the vision encoder. We train our VaCo on LLavA-558K and

Table 4: **Ablation Study** for (a) visual activation strategies and (b) various aligned VFMs. We investigate several visual activation strategies: (i) directly feeding the VFM's visual tokens into the LLM, and (ii) aligning the VFM's visual features with the MLLM's visual tokens or learnable queries. And we separately assess the effectiveness of generative (*Gen.*) and discriminative (*Dis.*) alignment objectives. Additionally, we examine the impact of the VFM choice on the performance of our method, alongside the role of TGM in coordinating visual features from multiple VFMs.

<table>
<tr><td colspan="7">(a) Ablation for Visual Activation Strategies</td><td colspan="7">(b) Ablation for Various VFMs</td></tr>
<tr><th>Method</th><th>Hallu.</th><th>MMBEN</th><th>SEEDI</th><th>MMMU</th><th>MMVP</th><th>Real.</th><th>Method</th><th>Hallu.</th><th>MMBEN</th><th>SEEDI</th><th>MMMU</th><th>MMVP</th><th>Real.</th></tr>
<tr><td>Baseline</td><td>47.5</td><td>65.5</td><td>66.0</td><td>34.5</td><td>20.0</td><td>52.7</td><td>w/ DINOv2</td><td>53.5</td><td>66.3</td><td>65.1</td><td>32.1</td><td>33.6</td><td>52.7</td></tr>
<tr><td>Token Input</td><td>50.3</td><td>65.6</td><td>65.4</td><td>33.2</td><td>22.5</td><td>52.8</td><td>w/ DAv2</td><td>53.5</td><td>66.5</td><td>66.0</td><td>33.0</td><td>35.9</td><td>54.1</td></tr>
<tr><td>Token *Gen.*</td><td>53.5</td><td>66.1</td><td>65.0</td><td>34.0</td><td>31.5</td><td>52.8</td><td>w/ Oneformer</td><td>53.8</td><td>65.0</td><td>64.1</td><td>32.8</td><td>34.0</td><td>55.7</td></tr>
<tr><td>Token *Dis.*</td><td>52.4</td><td>64.2</td><td>63.1</td><td>32.2</td><td>28.0</td><td>52.3</td><td>w/ VGGT</td><td>53.3</td><td>67.5</td><td>66.2</td><td>33.3</td><td>36.6</td><td>56.1</td></tr>
<tr><td>Query *Gen.*</td><td>56.0</td><td>66.9</td><td>65.8</td><td>35.6</td><td>35.7</td><td>55.2</td><td>VaCo w/o TGM</td><td>53.2</td><td>65.1</td><td>64.4</td><td>31.8</td><td>30.6</td><td>53.7</td></tr>
<tr><td>Query *Dis.*</td><td>**57.2**</td><td>**68.5**</td><td>**67.1**</td><td>**36.8**</td><td>**37.9**</td><td>**57.6**</td><td>VaCo w/ TGM</td><td>**57.2**</td><td>**68.5**</td><td>**67.1**</td><td>**36.8**</td><td>**37.9**</td><td>**57.6**</td></tr>
</table>

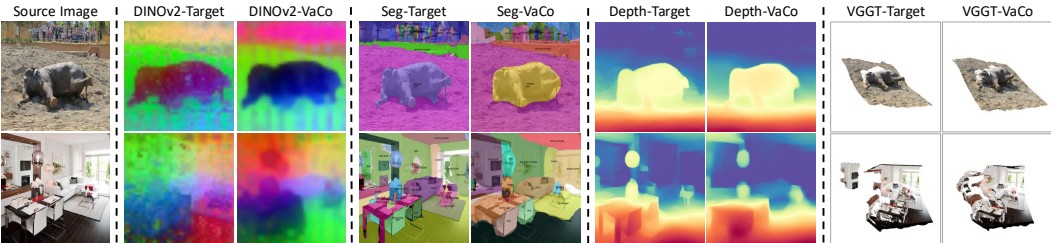

Figure 5: **Causal Perception Results** of our VaCo. For each tuple corresponding to the input original image, we present the output from the MLLM task queries and the output from the pure visual foundation model (right), including DINOv2 Oquab et al. (2023), DepthAnything V2 Yang et al. (2024a), OneFormer Jain et al. (2023), and VGGT Wang et al. (2025). The aligned perception results suggest that VaCo effectively pre-summarizes specific visual priors via task queries in visual understanding, thereby enhancing the correctness of text output through the causal mechanism.

Cambrian-737K datasets during the pre-training and the instruction tuning stages, respectively. All variants of VaCo are evaluated on (1) HaluBench Guan et al. (2024): Hall.; (2) MMBEN: MMBench-English Liu et al. (2024b); (3) SEEDI: SEED-Image Li et al. (2023a); (4) MMMU Yue et al. (2024); (5) MMVP Tong et al. (2024b); and (6) Real.: RealWorldQA x.ai (2024) benchmarks.

**Effect of Visual Activation Strategies**. As presented in Table 4 (a), we conduct an ablation study on different visual activation strategies: (1) directly inputting VFM tokens into the MLLM, and (2) reconstructing VFM features utilizing MLLM vision tokens or learnable MTQs, with a generative or discriminative objective. All non-baseline models adopt multiple VFMs, with TGM coordinating the query-based variants. We observe that both the generation and discrimination of visual signals outperform the direct integration method. Besides, as the query-based approach applies TGM to the causal mechanism of MLLM, it is more effective in avoiding feature conflicts among multiple VFMs compared to directly manipulating the vision tokens of MLLM. Note that ROSS Wang et al. (2024a) restricts its analysis to a generative/discriminative objective on the features of *single VFM*, using the vision tokens from MLLM. In contrast, our ablation studies show a more effective strategy: supervising multiple task-specific queries with a discriminative loss applied to the designed visual alignment layer. This setup better exploits multiple VFMs to activate visual signals within MLLMs.

**Effect of Various VFMs**. We further perform an ablation study to investigate the impact of single/multiple VFMs on the visual activation of MLLM, as well as the role of Token-Gated Mask (TGM) in coordinating various VFMs, as shown in Table 4 (b). Specifically, we first investigate the contributions of several methods to the performance of VaCo, including the self-supervised DINOv2 Oquab et al. (2023), the depth-estimation-scheme DepthAnything V2 Yang et al. (2024a), the segmentation-scheme OneFormer Jain et al. (2023), and the 3D-reconstruction-scheme VGGT Wang et al. (2025). The results indicate that VGGT enables MLLM to achieve superior visual reasoning performance, which suggests that insufficient multi-view 3D perception is the key bottleneck in limiting 2D image understanding in MLLMs. Meanwhile, we observe that directly integrating multiple task-specific MTQs into VaCo causes a notable performance drop, likely due to feature conflicts making the queries incompatible with the causal attention mechanism. The results demonstrate that our TGM effectively resolves this issue by coordinating VFMs through attention masking.

### 4.3 CAUSAL PERCEPTION RESULTS BY MLLM

Our VaCo coordinates different VFMs using multiple task queries, effectively activating various visual priors. Benefiting from this mechanism, VaCo performs visual understanding tasks without a substantial increase in computational demands, in contrast to Cambrian-1 Tong et al. (2024a) that employs multiple visual encoders. Despite perceptual results not being essential for visual understanding, integrating specific task queries with the Visual Alignment Layer allows the MLLM to obtain visual perceptual outcomes during causal inference. As shown in Figure 5, we present the causal perception results of MLLM, which validate the effectiveness of our VaCo in activating visual information. In other words, VaCo prompts the MLLM to distill specific visual features through diverse task queries before delivering textual output, ensuring that task queries and visual tokens together serve as a prerequisite for text output.

## 5 RELATED WORKS

### 5.1 MULTIMODAL LARGE LANGUAGE MODELS

Recent advances in LLMs Achiam et al. (2023); Chiang et al. (2023); Touvron et al. (2023); Zeng et al. (2022) have driven substantial progress in multimodal LLMs (MLLMs) for visual recognition and understanding. LLM-based multimodal models Liu et al. (2023a); Bai et al. (2023); Chen et al. (2023a; 2024); Wang et al. (2023b); Zhai et al. (2022); Zhu et al. (2023) aim to integrate the advanced understanding capabilities of LLMs with the visual information provided by language-supervised visual encoders Radford et al. (2021); Zellers et al. (2019); Zhai et al. (2023); Tong et al. (2024b). A connector Liu et al. (2023a; 2024a); Alayrac et al. (2022); Li et al. (2023b); Ge et al. (2023); Li et al. (2024a) is employed to convert visual signals from the visual encoder into visual tokens within the LLM representational space. Despite these models exhibiting powerful performance, the lack of explicit visual supervision inherently leads to the degradation of rich visual information. Recent studies Wang et al. (2024a); Huang et al. (2025) have emphasized that activating visual signals boosts the comprehension capabilities of MLLMs. In contrast to ROSS Wang et al. (2024a), which reconstructs *single* visual features from MLLM visual tokens, our VaCo adopts a query-based pipeline that enables more flexible alignment of task-aware perceptual priors across *multiple* VFMs.

### 5.2 VISION FOUNDATION MODELS IN MLLMS

The pioneering LLaVA Liu et al. (2023a) model adopts CLIP Radford et al. (2021) as its visual encoder, and subsequent work has integrated stronger vision–language alignment encoders Zhai et al. (2023); Chen et al. (2024) into MLLM architectures. Building upon these approaches, some models Tong et al. (2024a;b) incorporate multiple visual encoders, such as DINOv2 Oquab et al. (2023), aMAE He et al. (2022) and MoCoV3 He et al. (2020), alongside vision-language alignment models. In recent years, certain works Jain et al. (2024); Huang et al. (2025) have discovered that visual signals from task-specific models address the limitation of CLIP, which mainly captures global information. Consequently, task-specific models such as DepthAnything Yang et al. (2024a;b), OneFormer Jain et al. (2023), FLARE Zhang et al. (2025) and VGGT Wang et al. (2025) have potential for comprehensive integration into MLLMs. Nonetheless, multi-encoder schemes, such as Cambrain-1 Tong et al. (2024a) and VCoder Jain et al. (2024), directly feed multi-source features into MLLMs, increasing the visual encoding cost while leaving potential representation conflicts unaddressed. Unlike these works, our VaCo leverages learnable task-specific queries to *activate* and *coordinate* the knowledge from *multiple* VFMs, thereby enhancing the visual comprehension capabilities of MLLMs without significantly increasing computational overhead.

## 6 CONCLUSION

In this study, we address the limitations of current multimodal large language models (MLLMs) by emphasizing the integration of critical vision-centric information that enhances analytical capabilities. The innovation of our framework lies in supervising both textual and visual outputs through Vision-Centric Activation and Coordination (VaCo), harnessing comprehensive visual priors from multiple vision foundation models (VFMs). Through its unified framework, VaCo manages vision-centric representation activations, focusing on specific dimensions for improving visual understanding. This is achieved by learnable Modular Task Queries (MTQs), Visual Alignment Layers (VALs), and a Token Gateway Mask (TGM), facilitating effective communication between visual tokens and multiple VFMs. Extensive experiments demonstrate that our VaCo significantly improves the MLLM performance across diverse benchmarks, encompassing general, region-level, and scene-level understanding tasks, showcasing its advanced vision-comprehension capabilities.

## ETHICS STATEMENT

This research does not involve human subjects or the collection of personally identifiable information. All datasets utilized in this work are publicly available and comply with established ethical usage policies. We are aware that large language models and vision foundation models may have potential risks related to bias, fairness, and privacy, often inherited from underlying datasets. We urge caution when deploying such models in high-stakes scenarios, and recommend further rigorous assessments for bias, fairness, and safety in real-world applications. If any concerns arise regarding the ethical aspects of this paper, we are open to further discussion and clarification.

## REPRODUCIBILITY STATEMENT

We are committed to ensuring the reproducibility of our work. Comprehensive details regarding the training procedures, dataset descriptions, and evaluation protocols are provided in Section 4 of the *main paper* and Section A of the *Appendix*. To further facilitate reproducibility, we provide an anonymous source code package as part of the *Supplemental Materials*. We hope these resources enable the research community to easily reproduce our results and build upon our contributions.

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

APPENDIX

## A  IMPLEMENTATION DETAILS

### A.1  TRAINING DETAILS

**Training Setup**. Similar to LLaVA Liu et al. (2023a), the overall training process adopts a two-stage paradigm, comprising Pre-Training (PT) followed by Supervised Fine-Tuning (SFT). In Table 5, we outline the detailed two-stage training process of VaCo. The image resolution for CLIP-ViT-L/14@336 Radford et al. (2021) is set to $336 \times 336$, while that for SigLIP-ViT-SO400M/14@384 Zhai et al. (2023) is set to $384 \times 384$. The visual activation mechanism of VaCo can be seamlessly integrated into existing multimodal pipelines, as it only requires incorporating task-specific MTQs into the input of the LLM (with the additional VAL for visual alignment being necessary solely during training). The details of the selected VFMs are presented in Table 6.

Table 5: **Training Setup of VaCo**.

| Setting | | Stage 1
Pre-Training (PT) | Stage2
Supervised Fine-Tuning (SFT) |
|---|---|---|---|
| **Modules** | Vision Encoder | Frozen | Frozen |
| | Vision Foundation Models | Frozen | Frozen |
| | Large Language Model | Frozen | Trainable |
| | Projection | Trainable | Trainable |
| | Modular Task Queries | Trainable | Trainable |
| | Visual Alignment Layer | Trainable | Trainable |
| **Hyperparameters** | Batch Size | 256 | 128 |
| | Learning Rate | 2e-3 | 2e-5 |
| | Epoch | 1 | 3 |
| | Schedule | Warmup + Cosine decay | |
| | Warmup Ratio | 0.03 | |
| | Weight decay | 0 | |
| | Optimizer | AdamW | |
| | Precision | bf16 | |
| | Layer Numbers $L$ of VAL | 3 | |

Table 6: **Details of the selected VFMs**. The batch Size is denoted as $B$.

| Models | URL | Input Resolution | Feature Dimension |
|---|---|---|---|
| DINOv2 | https://huggingface.co/facebook/dinov2-giant | $224 \times 224$ | $B \times 256 \times 1024$ |
| DepthAnything V2 | https://huggingface.co/depth-anything/Depth-Anything-V2-Large | $336 \times 336$ | $B \times 576 \times 1024$ |
| OneFormer | https://huggingface.co/shi-labs/oneformer_coco_swin_large | $800 \times 800$ | $B \times 576 \times 1536$ |
| VGGT | https://huggingface.co/facebook/VGGT-1B | $518 \times 518$ | $B \times 1374 \times 2048$ |

**Datasets for Supervised Fine-Tuning**. Following ROSS Wang et al. (2024a), we utilize LLavA-558K Liu et al. (2023a) and Cambrian-737K Tong et al. (2024a) datasets during the Pre-Training (PT) and Supervised Fine-Tuning (SFT) stage, respectively. The composition details of Cambrian-737K are listed in Table 7 (a). In addition to general vision-language understanding, we also integrate the AS-V2 datasets Wang et al. (2024b) to demonstrate the capabilities of VaCo in improving region/scene-level understanding. The AS-V2 integrates the formulation of text generation, object localization, and relation comprehension into a relation conversation (ReC) task.

### A.2  EVALUATION DETAILS

**Benchmarks for Evaluation**. We perform a comprehensive evaluation of VaCo, covering general, region-level, and scene-level understanding benchmarks. Following recent works Li et al. (2024a); Wang et al. (2024a), we conduct the general vision-language understanding evaluation on 14 widely adopted benchmarks, which cover a diverse range of visual tasks. We provide a comprehensive overview of all evaluation benchmarks utilized in this paper in Table 7 (b). We employ the lmms-eval Bo et al. (2024) and VLMEvalKit Duan et al. (2024) toolboxes to conduct the majority of the evaluations presented in the main text and appendix, while following the Cambrian-1 Tong et al. (2024a) to evaluate performance on MMVP Tong et al. (2024b). For region-/scene-level evaluations, we adopt the settings from the ASMv2 Wang et al. (2024b).

Table 7: **Details of Datasets and Benchmarks**.

(a) Datasets for SFT stage.

| Dataset | #Samples |
|---|---|
| LLaVA Liu et al. (2023a) | 158K |
| ShareGPT ShareGPT Team (2023) | 40K |
| VQAv2 Goyal et al. (2017) | 83K |
| GQA Hudson & Manning (2019) | 72.1K |
| OKVQA Marino et al. (2019) | 9K |
| OCRVQA Marino et al. (2019) | 80K |
| A-OKVQA Schwenk et al. (2022) | 50K |
| TextVQA Singh et al. (2019) | 21.9K |
| RefCOCO Kazemzadeh et al. (2014) | 30K |
| VG Krishna et al. (2017) | 86.4K |
| DVQA Kafle et al. (2018) | 13K |
| DocVQA Mathew et al. (2021) | 15K |
| ChartQA Masry et al. (2022) | 28.1K |
| AI2 Diagrams Kembhavi et al. (2016) | 15.5K |

(b) Benchmarks for evaluation

| Benchmark | Response Format |
|---|---|
| POPE Li et al. (2023c) | - |
| HallusionBench Guan et al. (2024) | A single choice |
| MMBench Liu et al. (2024b) | A single choice |
| SEED-Bench Li et al. (2023a) | A single choice |
| MMMU Yue et al. (2024) | A single choice |
| MMVP Tong et al. (2024b) | A single choice |
| AI2D Hiippala et al. (2021) | A single choice |
| RealWorldQA x.ai (2024) | A single choice |
| GQA Hudson & Manning (2019) | A single word or phrase |
| ChartQA Masry et al. (2022) | A single word or phrase |
| OCRBench Liu et al. (2023b) | A single word or phrase |
| DocVQA Mathew et al. (2021) | A single word or phrase |
| InfoVQA Biten et al. (2022) | A single word or phrase |
| TextVQA Singh et al. (2019) | A single word or phrase |

Table 8: **Comparison of Computational Cost**.

| Method | #Vision Encoders | #Vision Tokens | #Task Queries | FLOPs (T) Vision Encoder | Projection | LLM | Total | Params (B) | Time (ms) |
|---|---|---|---|---|---|---|---|---|---|
| LLaVA-v1.5 Liu et al. (2023a) | 1 | 576 | 0 | 0.349 | 0.024 | 8.177 | 8.55 | 7.06 | 135.5 |
| ROSS Wang et al. (2024a) | 1 | 576 | 0 | 0.349 | 0.024 | 8.177 | 8.55 | 7.06 | 135.5 |
| Cambrian-1 Tong et al. (2024a) | 4 | 576 | 0 | 4.523 | 0.298 | 8.177 | 13.0 | 8.58 | 192.7 |
| VaCo (**ours**) | 1 | 576 | 1 | 0.349 | 0.024 | 8.224 | 8.60 | 7.06 | 135.9 |
| VaCo (**ours**) | 1 | 576 | 4 | 0.349 | 0.024 | 8.385 | 8.76 | 7.06 | 138.6 |
| VaCo (**ours**) | 1 | 576 | 8 | 0.349 | 0.024 | 8.591 | 8.96 | 7.06 | 140.4 |
| VaCo (**ours**) | 1 | 576 | 16 | 0.349 | 0.024 | 9.012 | 9.39 | 7.06 | 146.8 |

Table 9: **Ablations on Choices of Different LLMs and Visual Encoders**.

| Benchmark | CLIP-ViT-L/14@336 Vicuna-7B-v1.5 | | Qwen2-7B-Instruct | | SigLIP-ViT-SO400M/14@384 Vicuna-7B-v1.5 | | Qwen2-7B-Instruct | |
|---|---|---|---|---|---|---|---|---|
| | LLaVA | VaCo | LLaVA | VaCo | LLaVA | VaCo | LLaVA | VaCo |
| POPE | 86.3 | 87.0 (+**0.7**) | 87.9 | 88.4 (+**0.5**) | 86.0 | 87.4 (+**1.4**) | 88.5 | 88.4 (−**0.1**) |
| HallusionBench | 52.5 | 57.2 (+**4.7**) | 55.0 | 58.8 (+**3.8**) | 50.4 | 53.9 (+**3.5**) | 57.3 | 57.9 (+**0.6**) |
| MMBench-EN | 67.0 | 68.5 (+**1.5**) | 73.8 | 77.6 (+**3.8**) | 64.5 | 73.3 (+**8.8**) | 76.3 | 78.6 (+**2.3**) |
| MMBench-CN | 60.0 | 64.2 (+**4.2**) | 72.9 | 74.8 (+**1.9**) | 63.1 | 65.6 (+**2.5**) | 75.7 | 76.9 (+**1.2**) |
| SEED-I | 66.7 | 67.1 (+**0.4**) | 70.3 | 71.7 (+**1.4**) | 68.2 | 70.1 (+**1.9**) | 72.3 | 73.0 (+**0.7**) |
| MMMU | 35.3 | 36.8 (+**1.5**) | 41.9 | 41.6 (−**0.3**) | 34.2 | 35.4 (+**1.2**) | 41.8 | 46.7 (+**4.9**) |
| MMVP | 28.0 | 37.9 (+**9.9**) | 29.3 | 45.6 (+**16.3**) | 27.3 | 41.2 (+**13.9**) | 40.7 | 52.7 (+**12.0**) |
| AI2D | 61.2 | 68.3 (+**7.1**) | 71.9 | 77.7 (+**5.8**) | 62.6 | 66.6 (+**4.0**) | 74.0 | 78.9 (+**4.9**) |
| ChartQA | 32.9 | 42.2 (+**9.3**) | 36.2 | 44.5 (+**8.3**) | 34.0 | 50.6 (+**16.6**) | 44.4 | 49.3 (+**4.9**) |
| DocVQA | 33.4 | 42.6 (+**9.2**) | 31.1 | 45.9 (+**14.8**) | 40.4 | 40.4 (+**0.0**) | 39.2 | 41.3 (+**2.1**) |
| InfoVQA | 21.2 | 28.4 (+**7.2**) | 22.1 | 42.8 (+**20.7**) | 22.8 | 26.6 (+**3.8**) | 24.0 | 27.9 (+**3.9**) |
| TextVQA | 55.7 | 61.3 (+**5.6**) | 52.0 | 58.8 (+**6.8**) | 60.5 | 64.6 (+**4.1**) | 56.3 | 60.9 (+**4.6**) |
| OCRBench | 339 | 370 (+**31**) | 363 | 421 (+**58**) | 354 | 397 (+**43**) | 432 | 462 (+**30**) |
| RealWorldQA | 52.7 | 57.6 (+**4.9**) | 56.7 | 61.6 (+**4.9**) | 55.0 | 61.5 (+**6.5**) | 57.9 | 63.7 (+**5.8**) |
| Average | 70.85 | 77.74 (+**6.89**) | 76.01 | 86.49 (+**10.48**) | 73.07 | 81.01 (+**7.94**) | 84.31 | 89.87 (+**5.56**) |

# B MORE EXPERIMENTS

## B.1 COMPARISON OF COMPUTATIONAL COST

In Table 8, we present the computational cost results compared to the reconstructive Wang et al. (2024a) and multi-encoder Tong et al. (2024a) schemes. We investigate the FLOPs of each module and the total parameter count, while testing the inference latency on a single NVIDIA H20 GPU. We adopt LLaVA-v1.5 Liu et al.

Table 10: **Ablations on Number of MTQs**. The best results are in **red**, and the second-best are in **green**.

| Method | Hallu. | MMB[EN] | SEED[I] | MMMU | MMVP | Real. |
|---|---|---|---|---|---|---|
| VaCo ($Q = 1$) | 56.5 | 66.9 | 66.5 | 35.8 | 35.3 | 55.6 |
| VaCo ($Q = 4$) | **57.1** | 67.8 | **66.6** | 35.5 | **36.6** | 56.7 |
| VaCo ($Q = 8$) | **57.2** | **68.5** | **67.1** | **36.8** | **37.9** | **57.6** |
| VaCo ($Q = 16$) | **57.1** | **68.5** | **67.1** | **37.0** | 36.4 | **57.2** |

(2023a) as the baseline, which employs Vicuna-7B Chiang et al. (2023) as the LLM backbone and CLIP-ViT-L/14@336 Radford et al. (2021) as the vision encoder. For a fair comparison, all the models are built on the same architecture. As mentioned in the main paper, we implement 4 visual encoders for the multi-encoder scheme and our VaCo. As an intrinsic activation method, ROSS elim-

Table 11: **Quantitative Results of Affine-Invariant Monocular Depth Estimation**. We conduct our evaluation pipeline on NYUv2 Silberman et al. (2012), KITTI Geiger et al. (2012), ETH3D Schops et al. (2019) and DIODE Vasiljevic et al. (2019) datasets.

| Method | NYUv2 | | KITTI | | ETH3D | | DIODE | |
|---|---|---|---|---|---|---|---|---|
| | Rel↓ | $\delta_1\uparrow$ | Rel↓ | $\delta_1\uparrow$ | Rel↓ | $\delta_1\uparrow$ | Rel↓ | $\delta_1\uparrow$ |
| Depth Anything V2 Yang et al. (2024a) | 4.4 | 97.9 | 7.5 | 94.8 | 13.2 | 86.2 | 6.5 | 95.4 |
| VaCo (**ours**) | 6.9 | 90.2 | 11.8 | 85.1 | 18.7 | 77.6 | 10.3 | 86.0 |
| VGGT (Depth+Cam) Wang et al. (2025) | 3.6 | 98.0 | 9.3 | 91.7 | 4.1 | 97.2 | 27.4 | 78.7 |
| VaCo (**ours**) | 6.7 | 89.9 | 12.8 | 80.5 | 9.6 | 86.7 | 33.4 | 69.2 |
| VGGT (Point) Wang et al. (2025) | 3.6 | 98.0 | 9.5 | 91.5 | 3.9 | 97.7 | 27.8 | 78.6 |
| VaCo (**ours**) | 6.6 | 89.9 | 12.9 | 80.1 | 9.3 | 87.3 | 34.3 | 68.1 |

inates additional computational overhead during inference but *prioritizes optimizing reconstruction quality* over accurately capturing true semantics. In contrast, although the additional MTQs of our VaCo slightly increase the computational load (about $+4.7\%$ compared to baseline when $Q = 8$), it accurately activates the specific visual signal of the MLLMs (as shown in Figure 1) and promotes the optimization of the textual loss (as shown in Figure 2). Furthermore, Cambrian-1 takes features from multiple visual encoders as MLLM input during inference and utilizes a more complex visual projector. This results in two major issues: (a) *increased computational demands* and (b) *significant alignment burdens*. Alternatively, our VaCo employs lightweight MTQs during inference instead of multiple visual encoders, substantially reducing computational overhead (about $-31.1\%$ compared to Cambrian-1 when $Q = 8$) while delivering improved performance.

### B.2 MORE ABLATION STUDIES

**Ablations on Choices of Different LLMs and Visual Encoders**. In Table 9, we perform experiments utilizing diverse LLM backbones and visual encoders, which further evaluate the effectiveness of our vision-centric activation and coordination approach. Building upon Table 1, we include additional benchmarks in Table 9, demonstrating that different variants of VaCo consistently deliver significant improvements over the baseline. This confirms that VaCo facilitates flexible integration with existing LLM frameworks, while also allowing for the independent selection of visual encoders. Furthermore, the results indicate that the VaCo variants exhibit more prominent improvements on benchmarks focused on fine-grained visual understanding, such as MMVP Tong et al. (2024b). This further confirms the capability of VaCo to effectively utilize diverse VFMs to activate visual priors.

**Ablations on Number of Task Queries (MTQs)**. In Table 10, we examine the impact of the MTQ quantity $Q$ allocated to each task on the performance of VaCo. Theoretically, increasing the number of queries improves the representation capability for specific visual priors, but it also incurs higher computational costs (as illustrated in Table 8). From the results in Table 8, we observe that increasing the number of MTQs initially (from 1 to 8) brings significant performance gains. However, once the query quantity reaches a certain threshold ($Q = 16$), its effect on the performance becomes negligible. This is likely because a properly sized set of queries is sufficient to capture the activated visual representations from VFMs, whereas a larger number fails to provide additional information and could even introduce redundancy. Therefore, to balance performance and computational costs, we selected $Q = 8$ as the basic configuration for VaCo in the comparative experiments.

**Ablations on Latest VFMs.** As DINOv3 Siméoni et al. (2025) has been recently released and represents the latest advancement in visual foundation models (VFMs), we conducted a supplementary

Table 12: **Ablations on DINOv3** Siméoni et al. (2025).

| Method | Hallu. | MMB$^{EN}$ | SEED$^I$ | MMMU | MMVP | Real. |
|---|---|---|---|---|---|---|
| w/ DINOv2 | **53.5** | 66.3 | 65.1 | 32.1 | 33.6 | 52.7 |
| w/ DINOv3 | 53.3 | **67.3** | **66.0** | **33.5** | **36.6** | **54.8** |

experiment to integrate a single DINOv3[*] into our framework. This experiment aims to demonstrate the flexibility of our method in adapting to state-of-the-art VFMs, while also providing a preliminary assessment of the potential performance gains when substituting DINOv2 with DINOv3. In Table 12 and Figure 10, we observed a consistent improvement across most evaluation benchmarks when replacing DINOv2 with DINOv3. These quantitative and qualitative results both suggest that DINOv3 can seamlessly integrate into our framework and potentially enhance performance.

---

[*]https://huggingface.co/facebook/dinov3-vith16plus-pretrain-lvd1689m

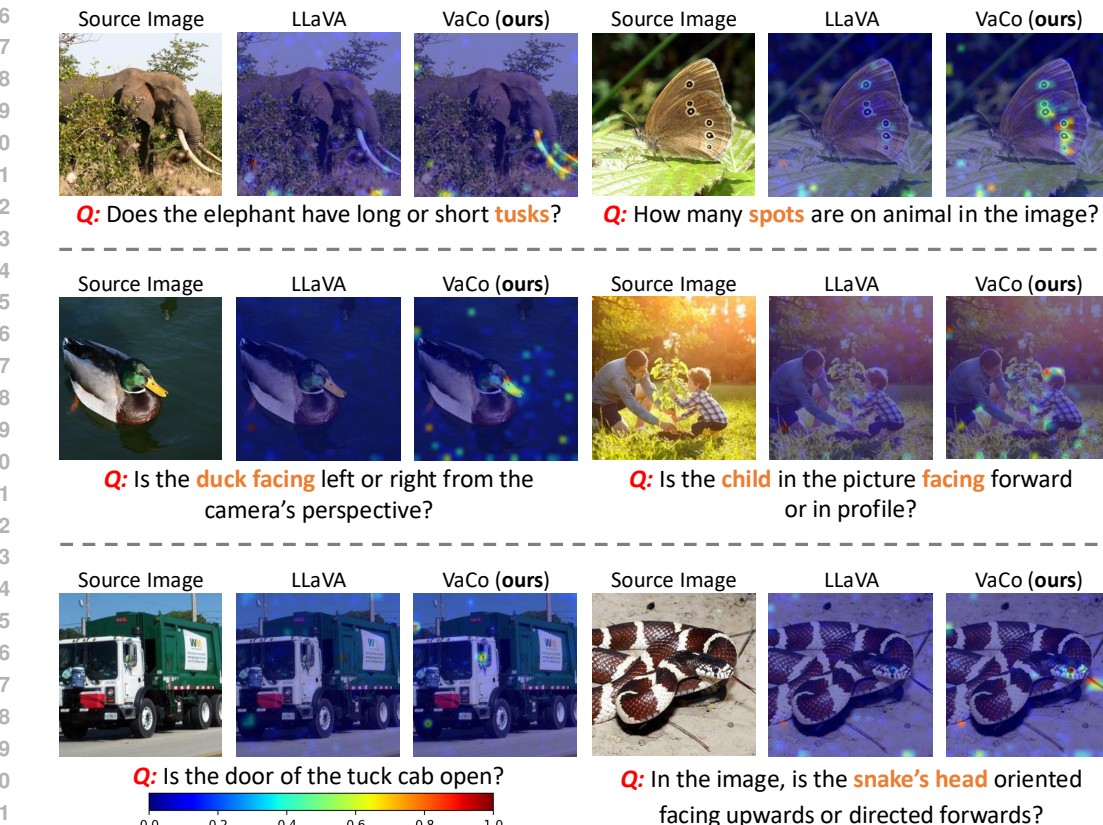

Figure 6: **Visualization of the Attention Scores**. The left side shows the input image, while the middle and right sides display the attention visualization results of LLaVA and VaCo, respectively. The example images and questions are all sourced from the MMVP Tong et al. (2024b) benchmark.

### B.3 QUANTITATIVE PERCEPTUAL RESULTS

As we mentioned in the main text, MTQ first reconstructs the latent representation of a specific visual task in the causal process, ultimately promoting the text-output understanding of MLLMs. While the VAL, responsible for visual output, is not essential for visual understanding during

Table 13: **Quantitative Results of Segmentation** on COCO Lin et al. (2014) val2017 set.

| Method | PQ | PQ$^{thing}$ | PQ$^{stuff}$ | AP | mIoU |
|---|---|---|---|---|---|
| OneFormer Jain et al. (2023) | 57.9 | 64.4 | 48.0 | 49.0 | 67.4 |
| VaCo (**ours**) | 43.9 | 48.2 | 33.7 | 37.6 | 53.3 |

inference, it can still be combined with MTQs to yield perceptual results as an additional byproduct, as shown in Figure 5. To quantitatively assess the representation capability of MTQ for specific visual tasks, we report the quantitative metrics of the visual results output by VAL across various perception tasks in Tables 11 and 13. For evaluation metrics of depth estimation Yang et al. (2024b), we use the relative point error $Rel$ and the percentage of inliers $\delta_1$. And we report the PQ Kirillov et al. (2019), AP Lin et al. (2014), and mIoU Everingham et al. (2015) scores for segmentation task.

### B.4 MORE QUALITATIVE RESULTS

**Visualization of the Attention Scores**. Similar to Figure 1, we present additional visualizations of attention scores in Figure 6. In particular, as MLLMs are designed to generate answers at the end of the input sequence, we visualize the attention scores between the final token and all image tokens. VaCo proposes a query-driven activation strategy that adaptively activates vision tokens via existing VFMs while retaining essential information. The learnable MTQs interact with all vision tokens through causal attention to effectively capture key visual details. Across various question types (*e.g.*, location, count, and tendency), the activation of VaCo demonstrates strong interpretability, effectively concentrating on the critical visual information within images. Overall, compared to the LLaVA baseline, the query-based VaCo adaptively assigns greater emphasis to key information, effectively activating critical visual details essential for visual understanding.

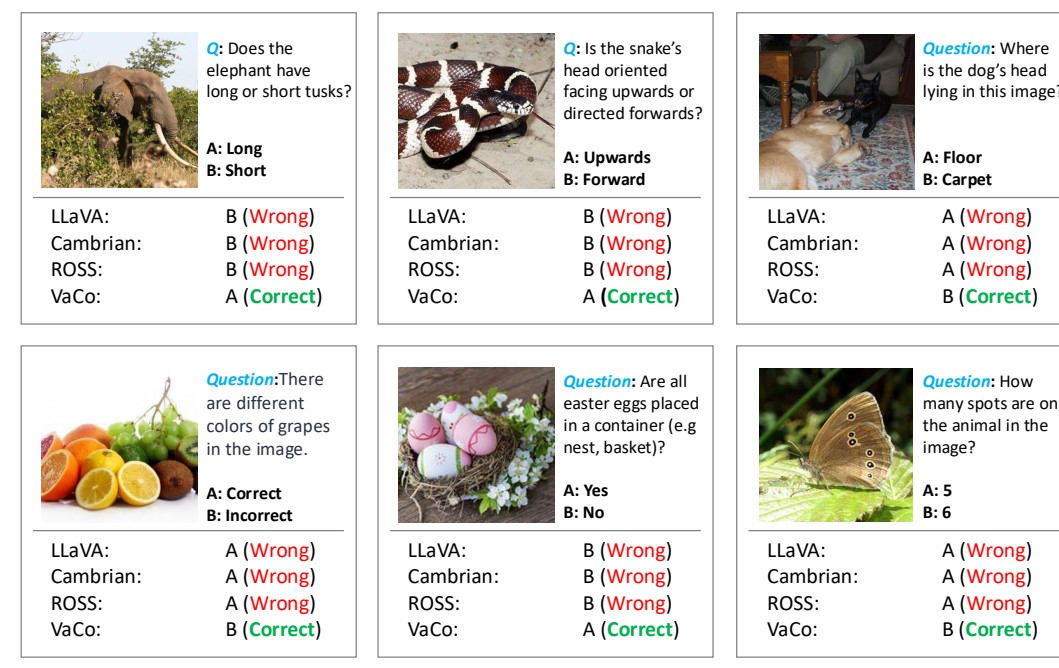

Figure 7: **Qualitative Cases on MMVP Tong et al. (2024b) Benchmark**.

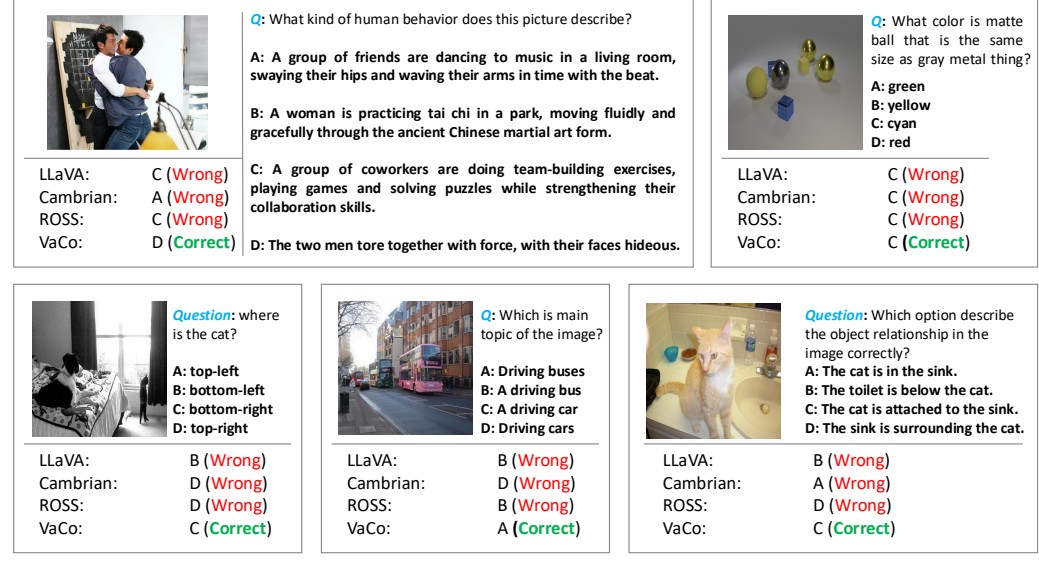

Figure 8: **Qualitative Cases on MMBench Liu et al. (2024b) Benchmark**.

**Image Understanding Cases**. To highlight the advantages of our VaCo over other visual activation approaches, we present extensive qualitative comparisons in Figure 7 and Figure 8, showcasing results on MMVP Tong et al. (2024b) and MMBench Liu et al. (2024b), respectively. We compare our approach with the baseline LLaVA Liu et al. (2023a), the reconstructive model ROSS Wang et al. (2024a), and the multi-encoder framework Cambrian-1 Tong et al. (2024a). In Figure 7, the qualitative results highlight the enhanced visual understanding capabilities (examples 1, 2, and 3), spatial localization ability (examples 4 and 5), and object counting skills (example 6) of our VaCo. From the top-left to the bottom-right in Figure 8, our VaCo demonstrates strong capabilities in understanding diverse aspects, such as action recognition, attribute recognition, object localization, image topics, and spatial relationships. This suggests that the introduced vision-centric activation effectively addresses the visual shortcomings of the original representation in MLLMs.

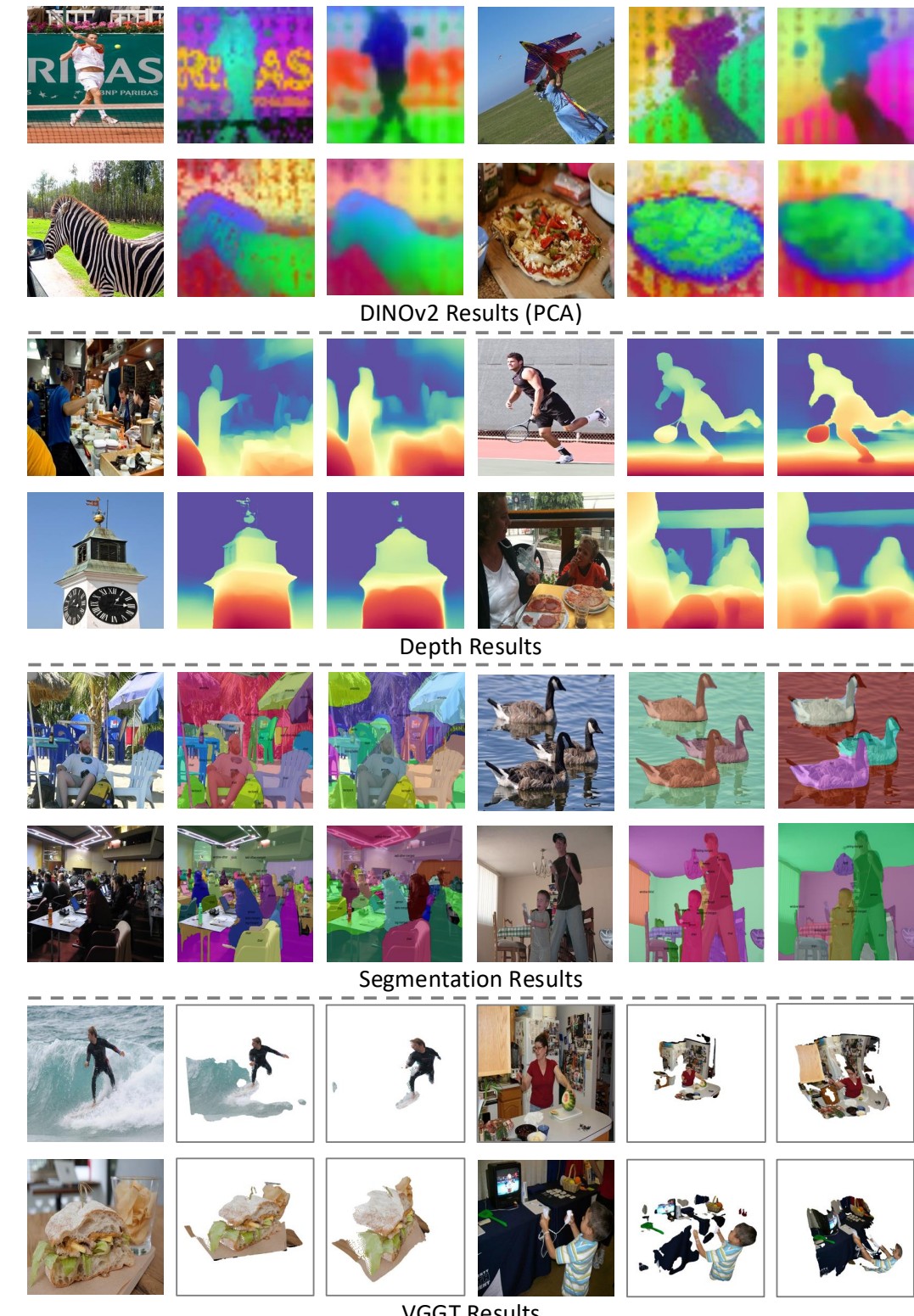

DINOv2 Results (PCA)

Depth Results

Segmentation Results

VGGT Results

Figure 9: **Perceptual Visualization by our VaCo**. For each example, the original image, VFM output, and VaCo output are displayed from left to right. This is a by-product of the Visual Alignment Layer (VAL), which is *not essential* for language-output understanding tasks.

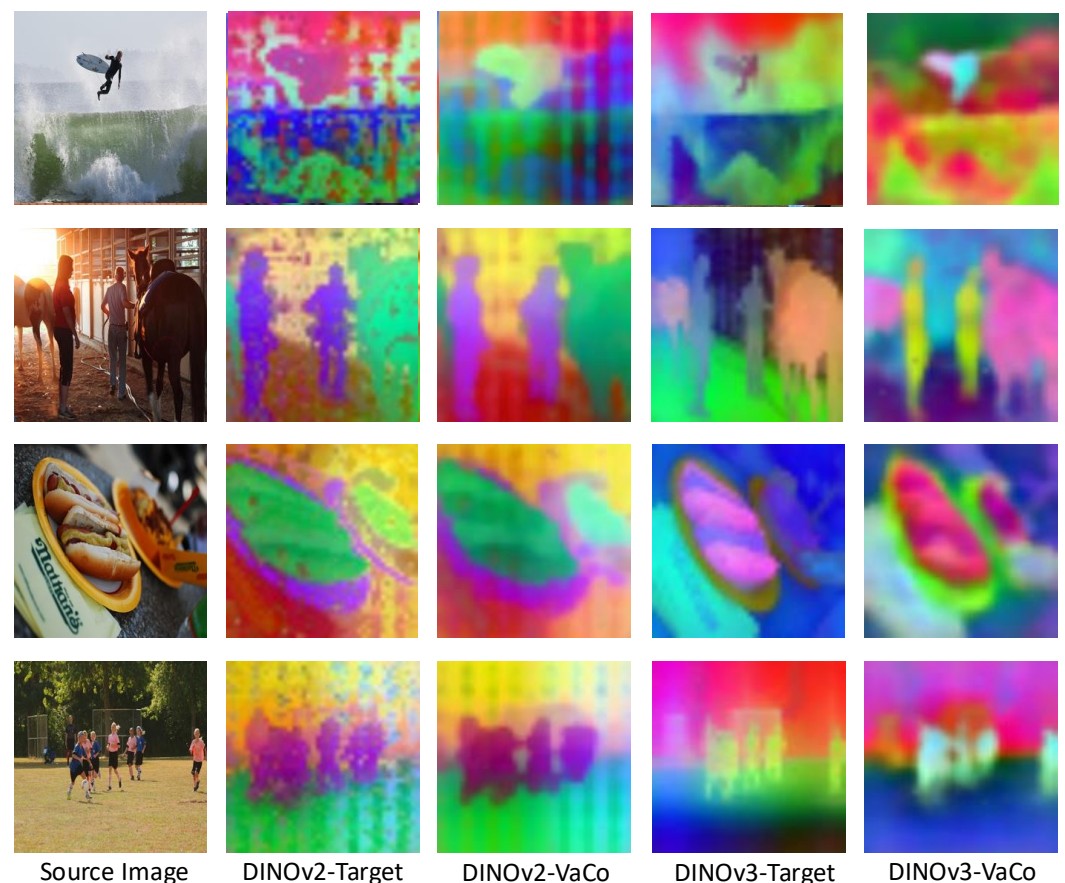

| Source Image | DINOv2-Target | DINOv2-VaCo | DINOv3-Target | DINOv3-VaCo |

Figure 10: **Visualization Comparison Between DINOv2 and DINOv3**. This is a by-product of the Visual Alignment Layer (VAL), which is *not essential* for language-output understanding tasks.

**Perceptual Visualization**. In Figure 9 and Figure 10, we present more byproduct visual results generated by VaCo through MTQs and VALs. As mentioned in Section 4.3 of the main paper, while this process is not essential for visual-language understanding, it showcases how VaCo facilitates the distillation of visual priors from various VFMs, effectively activating intrinsic visual signals.

## C LIMITATIONS

One limitation of our current pipeline lies in its reliance on activating or refining the visual tokens from the pre-trained text-image aligned encoder (*e.g.*, CLIP and SigLIP) of the MLLM. Consequently, the performance of our method is inherently constrained by the capabilities of the text-image aligned encoder, particularly in capturing fine-grained visual features. This dependency may introduce biases stemming from the pre-trained visual features, potentially impacting the ability of the model to fully comprehend nuanced details in multi-modal tasks. In future work, we plan to integrate our VaCo into encoder-free architectures, enabling a more flexible and unbiased learning paradigm that avoids the constraints imposed by the predefined text-image aligned encoder.

## D LLM USAGE

We acknowledge the use of a large language model (LLM) as a general-purpose writing assistance tool for this work. Specifically, the LLM was **solely employed to help with linguistic refinement and grammar correction** during the manuscript preparation. *It did not contribute to the formulation of the core research ideas, experimental design, or analysis presented in this paper*.

# E    REBUTTAL

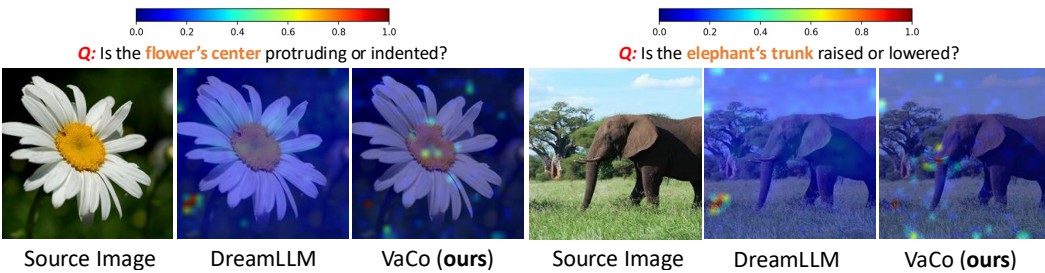

Figure 11: **Attention Visualization** of DreamLLM and VaCo. Utilizing samples from MMVP Tong et al. (2024b), we investigate which visual regions dominate the final predictions of MLLMs. Specifically, since MLLMs are anticipated to deliver answers at the final position of the input sequence, we visualize the attention scores between the last token and all image tokens. ***Compared to Dream-LLM, our VaCo effectively directs MLLMs to focus on the crucial areas mentioned in questions, demonstrating its effectiveness of integrating multiple visual encoders for visual understanding.***

| Method | MMB$^{EN}$ | MMVP | SEED$^I$ | Real. | MMMU |
|---|---|---|---|---|---|
| MSE | 78.0 | 50.4 | 72.8 | 62.2 | 45.5 |
| MSE+Contrastive | **78.6** | **52.7** | **73.0** | **63.7** | **46.7** |

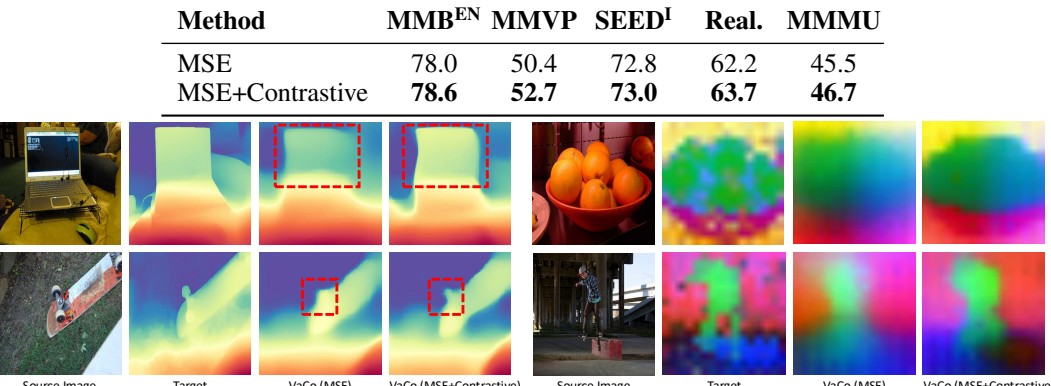

Figure 12: **Quantitative Results and Perceptual Examples with Different Losses**. We adopt SigLIP+Qwen2-7B to conduct ablations. It can be seen that results solely with MSE loss produce blurry predictions, whereas the addition of a contrastive term yields more structured and discriminative results. ***The improved perceptual quality leads to a consistent boost in understanding, thereby confirming the VaCo efficiency of integrating visual features for understanding.***

