# OpenReview forum: "Vision-Centric Activation and Coordination for Multimodal Large Language Models"
_ICLR.cc/2026/Conference — Submitted to ICLR 2026_

### Official Review · Reviewer_LW6u · 2025-10-21

**Soundness:** 3
**Presentation:** 3
**Contribution:** 3
**Rating:** 6
**Confidence:** 3

**Summary:**

The paper introduces VaCo, a method to enhance MLLM representations via Vision-Centric activation and coordination from multiple Vision Foundation Models (VFMs). VaCo uses Mechanism-specific Tunable Queries (MTQs) and Visual Activation Layers (VALs) to activate targeted visual signals under VFMs' supervision, incorporating a Tunable Gating Mechanism (TGM) to control information flow among MTQ groups.

**Strengths:**

1. Unlike previous works that introduced multiple visual encoders, this paper demonstrates increased efficiency during inference.
2. The authors have conducted extensive experiments to demonstrate the effectiveness of the proposed method, and VaCo significantly improves the performance of different MLLMs on various benchmarks

**Weaknesses:**

1. As shown in Table 1, VaCo consistently underperforms on the POPE dataset. The authors are requested to provide a plausible explanation for this result.
2. The model architecture used in the experiments of this paper is outdated. The authors are advised to validate the effectiveness of the proposed method using a more recent architecture(e.g., Qwen2.5-VL, Qwen3+SigLIP2) to strengthen the credibility of their method.
3. How does the training cost of VeCo compare to that of ROSE? Please provide a detailed comparison.
4. Why did you choose to use MSE + Contrastive loss during the training of the Visual Alignment Layer?

**Questions:**

See the above weaknesses.

---

> ### Author Response · Authors · 2025-11-21
> **Response to Reviewer LW6u**
>
> **Q1: Clarification for POPE scores**
>
> **R1**:  We agree that POPE is a valuable benchmark for evaluating object hallucination, which is why we included it in Table 1:
>
> + **Object hallucination, **_**under the setting of POPE**_**, has been largely resolved**. Recent methods [1,2] have attained POPE scores that are relatively close. In the table below, we present the results for methods that share the same LLM backbone, yet utilize more data [1, 2] or a stronger vision encoder [1] than VaCo:
>
> | **Method** | **Data Scale** | **Vision Encoder** | **LLM** | **POPE** |
> | --- | :---: | :---: | :---: | :---: |
> | Qwen2-VL-7B [1] | Private | QwenViT | Qwen2-7B | 88.4 |
> | LLaVA-OneVision-7B [2] | 4.8M | SigLIP-400M | Qwen2-7B | 88.4 |
> | VaCo-7B (ours) | 1.3M | SigLIP-400M | Qwen2-7B | 88.4 |
>
> Thus, POPE remains a useful benchmark for evaluating object hallucination, but it is _no longer highly discriminative among these advanced models_.
>
> + **HallusionBench [3] provides a more comprehensive evaluation of hallucination**. In Table 1, VaCo consistently outperforms the baselines on the more challenging HallusionBench, which assesses both language and visual hallucinations, highlighting the impact of visual features on mitigating hallucinations.
>
> **Q2: Effectiveness for Recent Architecture**
>
> **R2**: In the table below, we further validate the effectiveness of our method on more recent architectures, including _Qwen2.5+SigLIP_ and _Qwen3+SigLIP2_ on _Cambrian-737K_ dataset, to strengthen the credibility of our approach.
>
> | **Method** | **Vision Encoder** | **LLM** | **MMB_EN** | **MMVP** | **SEED-I** | **Real.** | **MMMU** |
> | --- | --- | --- | :---: | :---: | :---: | :---: | :---: |
> | LLaVA | SigLIP | Qwen2.5-7B | 76.9 | 48.6 | 74.5 | 58.6 | 50.1 |
> | VaCo | SigLIP | Qwen2.5-7B | **80.4** | **52.8** | **75.0** | **63.4** | **50.6** |
> | LLaVA | SigLIP2 | Qwen3-4B | 77.5 | 50.6 | 74.9 | 60.0 | 49.3 |
> | VaCo | SigLIP2 | Qwen3-4B | **80.8** | **54.4** | **76.2** | **64.2** | **54.1** |
>
> **Q3: Training Cost**
>
> **R3**:  In the table below, we present the performance and detailed training cost compared to ROSS. We conduct the architecture with _SigLIP+Qwen2-7B_ on the _Cambrian-737K_, using 8 H20 GPUs of 96GB memory. Since the vision models are frozen during training, we extract _offline visual features_ for efficient training. VaCo delivers _**significant performance gains without incurring excessive training costs**_:
>
> |  | **Training Cost** | | | | **Performance** | | | | | |
> | --- | :---: | :---: | :---: | :---: | :---: | :---: | :---: | :---: | :---: | :---: |
> | **Method** | **Trainable Params (B)** | **Speed (s/iter)** | **Time (h)** | **GPU Memory** | **MMB_EN** | **MMVP** | **SEED-I** | **Real.** | **MMMU** | **Average** |
> | ROSS | 7.696 | 13.56 | ~20 | 60.6 GB | 76.9 | 49.3 | 72.1 | 59.1 | 43.8 | 60.2 |
> | VaCo | 7.875 | 13.65 | ~20 | 61.5 GB | **78.6** | **52.7** | **73.0** | **63.7** | **46.7** | **62.9** |
>
> **Q4: Loss Selection**
>
> **R4**: We would clarify that the loss selection serves only as a training strategy, aiming to enhance the alignment between modalities rather than our main contribution. We apply MSE for precise local alignment with visual features, which is a common practice in distillation [4].  Motivated by previous work [5], we also add a contrastive loss for more discriminative and structured alignment. In the table below, we adopted _SigLIP+Qwen2-7B_ to ablate the effectiveness of these losses:
>
> | **Loss** | **MMB_EN** | **MMVP** | **SEED-I** | **Real.** | **MMMU** |
> | --- | :---: | :---: | :---: | :---: | :---: |
> | MSE | 78.0 | 50.4 | 72.8 | 62.2 | 45.5 |
> | MSE + Contrastive | **78.6** | **52.7** | **73.0** | **63.7** | **46.7** |
>
> Furthermore, in Figure 12 of the **Appendix**, the MSE results produce blurry predictions, whereas the addition of a contrastive term yields more structured and discriminative results. The better-aligned perceptual results lead to a _**consistent boost**_ in understanding benchmarks, thereby confirming the VaCo _**efficiency of integrating visual features for understanding**_.
>
> [1] Wang, Peng, et al. "Qwen2-vl: Enhancing vision-language model's perception of the world at any resolution." arXiv preprint arXiv:2409.12191.
>
> [2] Li, Bo, et al. "Llava-onevision: Easy visual task transfer." arXiv preprint arXiv:2408.03326.
>
> [3] Guan, Tianrui, et al. "Hallusionbench: an advanced diagnostic suite for entangled language hallucination and visual illusion in large vision-language models." CVPR 2024.
>
> [4] Sun, Siqi, et al. "Patient knowledge distillation for bert model compression." EMNLP 2019.
>
> [5] Tian, Yonglong, et al. "Contrastive representation distillation." ICLR 2020.

---

> > ### Comment · Reviewer_LW6u · 2025-11-27
> >
> > Thanks for your response, which has addressed my concerns. Considering the comments from other reviewers as well as your reply, I'm inclined to maintain my current positive score. Good luck!

---

### Official Review · Reviewer_m47S · 2025-10-26

**Soundness:** 3
**Presentation:** 4
**Contribution:** 2
**Rating:** 6
**Confidence:** 3

**Summary:**

The paper presents Vision-Centric Activation and Coordination (VaCo), which aims to enhance Visual abilities of VLMs by applying a query-based discriminative alignment with various vision experts. Specifically, they insert task-specific learnable queries (MTQ) into the LLM and align those queries to outputs of multiple frozen vision foundation models (VFM) via a small Visual Alignment Layer (VAL). Experimental results, both qualitative and quantitative, demonstrate that VaCo indeed enhances the performance of VLMs on vision tasks, and directs attention correctly.

**Strengths:**

1. The paper is partially novel to me. Instead of trying to inject multi-expert feature into the VLM like the traditional Cambrian or Eagle, this paper uses contrastive loss to align query-based VLM features with expert features. This sounds more elegant, and also reduces compute overhead.
2. The technical contributions, MTQ and TGM, are theoretically favourable. I agree with the authors' design logic in these two modules.
3. Experiments are generally solid. Authors performed main experiments on various types of VLMs, and did ablation studies to prove the effectiveness of MTQ and VFM selection.
4. The representation is clear. The figures are informative, and the visualizations strongly support their claims.

**Weaknesses:**

1. The idea of introducing contrastive loss into VLMs with experts is partially limited in novelty. Since I am not a research expert in this field, I cannot fully approve or disapprove of the novelty. I will refer to other reviewers' comments on novelty to decide my final score.
2. The experiments are done on rather old models like Vicuna or Qwen2-VL. The claims could have been strengthened if strong contemporary methods like Qwen2.5-VL (5 months before submission) were included.
3. The VaCo features in Figure 5 can demonstrate that VaCo is learning something from experts, but the quality is far from satisfactory compared with the experts.
4. Typo: "VQAs" in the caption of Figure 3.

**Questions:**

1. Why is the quality of VaCo feature far from satisfactory compared with the experts? Given that VLMs have much larger parameters, they should not be much less competitive, given that you trained with a discriminative loss. I wish to know about possible reasons behind it, and how you plan to address it.
2. (Not a deficiency) What is the potential of VaCo of becoming a VLM generalist: can replace DINO, VGGT etc? What else are needed to achieve this goal?

I am planning to rate 7. I am giving a slightly lower score than I expect, but if other reviewers are confident that this idea is novel, and authors can address my concerns, I will raise my score.

---

> ### Author Response · Authors · 2025-11-21
> **Response to Reviewer m47S**
>
> **Q1: Clarity of Core Novelty**
>
> **R1**: The contrastive loss serves only as a training strategy, aiming to enhance the alignment between modalities rather than our main contribution. We would like to emphasize that our core novelty lies in _**efficiently integrating multiple visual encoders into multimodal understanding**_. Although recent works (e.g., [1]) have investigated multiple vision models as additional inputs, they suffer from clear drawbacks in inference efficiency, as shown in Table 8 of the **Appendix**.
>
> **Q2: Effectiveness for Qwen2.5 and Qwen3**
>
> **R2**: Thank you for this helpful suggestion regarding the baseline choice. In the table below, we further demonstrate the effectiveness of VaCo on more recent models, specifically _Qwen2.5-7B + SigLIP_ and _Qwen3-4B + SigLIP2_ on _Cambrian-737K_ dataset:
>
> | **Method** | **Vision Encoder** | **LLM** | **MMB_EN** | **MMVP** | **SEED-I** | **Real.** | **MMMU** |
> | --- | --- | --- | :---: | :---: | :---: | :---: | :---: |
> | LLaVA | SigLIP | Qwen2.5-7B | 76.9 | 48.6 | 74.5 | 58.6 | 50.1 |
> | VaCo | SigLIP | Qwen2.5-7B | **80.4** | **52.8** | **75.0** | **63.4** | **50.6** |
> | LLaVA | SigLIP2 | Qwen3-4B | 77.5 | 50.6 | 74.9 | 60.0 | 49.3 |
> | VaCo | SigLIP2 | Qwen3-4B | **80.8** | **54.4** | **76.2** | **64.2** | **54.1** |
>
> **Q3: Calcification for Perceptual Results**
>
> **R3**: We would like to clarify that our objective is to **_improve multi-modal understanding_** by efficiently integrating multiple visual features, rather than to perform these perceptual tasks with LLM. As noted in Figure 5, these results are byproducts of VaCo, which serve as qualitative evidence of alignment, rather than primary evaluation criteria:
>
> + _Effects of Perceptual Performance on Understanding_. Despite the perceptual quality gap between VaCo and experts, the core of VaCo lies in **the efficiency of integrating visual features for _enhanced understanding_**. In the table below, we demonstrate that the improved perceptual quality (Depth Anything V2)  with the increasing query number (from Q=1 to 8), leading to a _**consistent boost**_ in understanding, thereby confirming the VaCo (_Vicuna-7B+CLIP_) efficiency:
>
> |  | **Perceptual Quality** | | | **Understanding** | | | | | |
> | --- | :---: | :---: | :---: | :---: | :---: | :---: | :---: | :---: | :---: |
> | **Method** | **NYUv2 Rel ↓** | **NYUv2 δ ↑** | **COCO PQ ↑** | **MMB_EN** | **MMVP** | **SEED-I** | **Real.** | **MMMU** | **Average** |
> | Q=1 | 8.7 | 89.1 | 41.5 | 66.9 | 35.3 | 66.5 | 55.6 | 35.8 | 52.0 |
> | Q=8 | **6.9** | **90.2** | **43.9** | **68.5** | **37.9** | **67.1** | **57.6** | **36.8** | **53.6** |
>
> + _Larger Parameters of VLMs_.   As noted in the previous study [2], the autoregressive modeling of LLMs is _**more suited to discrete symbols rather than continuous visual space**_. Some studies [3, 4] have also revealed that the VLMs are still less competitive than specialist vision models for precise perception. Therefore, _despite the large parameters of VLM, dedicated visual decoders for continuous modeling are still retained in these studies_. By contrast, while VLMs lack competitiveness in fine-grained perception, the recent study [1] shows that perceptual sources can improve understanding, albeit with a higher computational cost than our approach.
>
> **Q4:  Future improvements**
>
> **R4**: VaCo reveals that VLMs can efficiently absorb valuable visual signals with minimal overhead, suggesting that VaCo-like mechanisms could advance VLMs to evolve into visual generalists:
>
> + **Mixture-of-Transformer (MoT) for both Discrete and Continuous Modeling**. MoT [5, 6] offers a potential pathway for transforming VaCo into a VLM generalist, complementing the autoregressive transformers (VLMs with powerful understanding abilities) and vision transformers (vision experts with strong perceptual abilities).
> + **Multi-Layer Alignment for Fine-grained Visual Representations**.  Recent studies [7] have shown that different LLM layers focus on distinct aspects. Therefore, multi-layer visual alignment can potentially enhance the visual capacity of LLM.
>
> [1] Tong, Peter, et al. "Cambrian-1: A fully open, vision-centric exploration of multimodal llms." NeurIPS 2024.
>
> [2] Chen, Ting, et al. "A unified sequence interface for vision tasks." NeurIPS 2022.
>
> [3] He, Junwen, et al. "Multi-modal instruction tuned llms with fine-grained visual perception." CVPR 2024.
>
> [4] Wu, Jialian, et al. "Grit: A generative region-to-text transformer for object understanding." ECCV 2024.
>
> [5] Liang, Weixin, et al. "Mixture-of-transformers: A sparse and scalable architecture for multi-modal foundation models." TMLR 2025.
>
> [6] Deng, Chaorui, et al. "Emerging properties in unified multimodal pretraining." arXiv preprint arXiv:2505.14683.
>
> [7] Zhang, Zhi, et al. "Cross-modal information flow in multimodal large language models." CVPR 2025.

---

> > ### Comment · Reviewer_m47S · 2025-11-25
> > **Experiments convincing; Explanation for Q3 acceptible; novelty unconvincing**
> >
> > The authors' new results on Q2 addressed my concern of generalization into new models. I recommend including this table in Main Paper if it gets accepted. Besides, the authors' explanation for Q3 "more suited to discrete symbols rather than continuous visual space" is convincing to me.
> >
> > However, the authors' answer for Q1 is not clear to me. The idea of using multiple encoders is far from a novel one, as the authors also cited the classical work of Cambrian-1; the components, MTQ, TGM are not impressively innovative but seem like incremental modular designs. I have reservations about the technical contribution.
> >
> > Based on these assessments, I would like to keep the scoring of 6.

---

> ### Author Response · Authors · 2025-11-26
> **Response to Reviewer m47S**
>
> **Q5: Comparison with the Multi-Encoder Scheme**
>
> **R5**: Thank you for your thoughtful response. We would like to clarify that our core insight lies in **_efficient output-level multiple supervision_ for intrinsic visual representations**, which is fundamentally different from the **_direct input-level integration_ of multiple visual features** in classical Cambrian-1 [1]. To address the _visual degradation_ and _inference computational drawback_ of multi-encoder schemes, we employ lightweight MTQs to _replace multiple vision models by distilling their knowledge_, supplemented by TGM for _stable multi-objective alignment_:
>
> - *Input-Level Integration vs. Output-Level Supervision*. As illustrated in Figure 1 of the main paper, direct input-level integration of multiple encoders hinders MLLMs from highly informative regions for understanding, thereby causing visual degradation. Thus, our primary technical contribution lies in demonstrating that **output-level visual supervision _facilitates visual understanding more effectively_ than simply integrating visual inputs**, as shown in the table below:
> | **Method** | **#Extra Encoders** | **MMB_EN** | **MMVP** | **SEED-I** | **Real.** | **MMMU** |
> | --- | :---: | :---: | :---: | :---: | :---: | :---: |
> | LLaVA (baseline) | 0 | 67.0 | 28.0 | 66.7 | 52.7 | 35.3 |
> | LLaVA + Input [1] | 4 | 67.5 | 30.5 | 66.4 | 55.4 | 35.1 |
> | VaCo (ours) | 4 | **68.5** | **37.9** | **67.1** | **57.6** | **36.8** |
>
> - *Efficiency of MTQ*. Distinct from Cambrian-1 [1], the MTQ offers superior improvements in visual understanding with _**lower inference latency (see table below)**_:
> | **Method** | **#Extra Encoders** | **#Vision Tokens** | **#Task Queries** | **FLOPs (T)** | **Params (B)** | **Time (ms)** |
> | --- | :---: | :---: | :---: | :---: | :---: | :---: |
> | LLaVA (baseline) | 0 | 576 | 0 | 8.55 | 7.06 | 135.5 |
> | Cambrian-1 | 4 | 576 | 0 | 13.0 | 8.58 | 192.7 |
> | VaCo (ours) | 4 | 576 | 1 | **8.60** | **7.06** | **135.9** |
> | VaCo (ours) | 4 | 576 | 4 | **8.76** | **7.06** | **138.6** |
> | VaCo (ours) | 4 | 576 | 8 | **8.96** | **7.06** | **140.4** |
>
> - *Masking Strategy for Multi-Objective Alignment*. Besides the efficiency bottleneck, the entangled integration of visual representations [1] hinders improvements in visual understanding. As shown in Table 4 (b), _**directly stacking multiple MTQs results in feature conflicts among the visual encoders**_. To align stably with multiple visual features, we adopt a masking strategy for disentangling MTQs within the autoregressive framework, facilitating multi-objective alignment for understanding.
>
> [1] Tong, Peter, et al. "Cambrian-1: A fully open, vision-centric exploration of multimodal llms." NeurIPS 2024.

---

### Official Review · Reviewer_P8bA · 2025-10-28

**Soundness:** 3
**Presentation:** 2
**Contribution:** 2
**Rating:** 4
**Confidence:** 4

**Summary:**

This paper introduces a method called VaCo (Vision-Centric Activation and Coordination) to enhance the visual understanding capabilities of Multimodal Large Language Models (MLLMs). VaCo leverages visual priors from multiple Vision Foundation Models (VFMs) to activate and coordinate visual representations within MLLMs. Specifically, the paper proposes Modular Task Queries (MTQs) and Visual Alignment Layers (VALs) to activate task-specific visual signals inside the MLLM, while a Token Gateway Mask (TGM) is used to resolve representation conflicts across different tasks. VaCo demonstrates strong performance across various vision-language benchmarks, validating its effectiveness.

**Strengths:**

1. The paper introduce some extra modules for improving the visual comprehension performance of MLLMs. e.g. Introducing MTQs and VALs to activates task-specific visual information within the MLLM, avoiding computational overhead caused by directly feeding multiple VFM features into the model. Introducing TGM effectively addresses representation conflicts among task-specific queries, improving model stability and performance.
2. The paper conducts extensive experiments on multiple benchmarks (e.g., MMBench, MMMU, RefCOCO, VCR) at general benchmarks as well as at region-level and scene-level tasks.

**Weaknesses:**

1. The paper is hard to follow. The paper's own work and the discussion of existing work are often mixed together, the introduction of the methods in this paper is often interrupted, which might hinder my understanding of the ingenuity of the modules proposed in the paper
2. It is expected that more training with more parameters and more data will lead to better performance. The paper didnot demonstrate that such performance improvement is more competitive than that of just scaling up under the framework presented in the paper.  Since the absolute performance definitely can't compare to some of the models on the leaderboards of the benchmark.

**Questions:**

1.  what is the computational cost during training? How to balance training scale and performance?
2.  There are some hyper-parameters, e.g. the length and structure of MTQs , how to chose ?

---

> ### Author Response · Authors · 2025-11-21
> **Response to Reviewer P8bA**
>
> **Q1: Clarity of the Paper Presentation**
>
> **R1**: In response to your suggestion, we have improved the methodological sections by separating the discussion of related work from our methods. _**The revised text is highlighted in red**_. We hope these modifications will enhance the readability and make the novelty of our modules more apparent.
>
> **Q2: Efficiency of Scaling Up and VaCo**
>
> **R2**:  To quantify the impact of parameters/data scaling, we have conducted extensive ablation studies:
>
> + **Parameter Scaling vs. VaCo**: We adopt LLaVA (_Vicuna-7B+CLIP_) trained on _Cambrian-737K_ as baseline, and investigate the effects of more parameters (_Vicuna-13B_) and VaCo. In the table below, VaCo surpasses parameter scaling (_from 7B to 13B_) on average and most individual metrics, demonstrating its superiority:
>
> | **Method** | **LLM** | **MMB_EN** | **MMVP** | **SEED-I** | **Real.** | **MMMU** | **Average** |
> | --- | --- | :---: | :---: | :---: | :---: | :---: | :---: |
> | LLaVA (baseline) | Vicuna-7B | 67.0 | 28.0 | 66.7 | 52.7 | 35.3 | 49.9 |
> | LLaVA | Vicuna-13B | **68.7** | 32.8 | **68.1** | 55.2 | 35.7 | 52.1 |
> | VaCo | Vicuna-7B | 68.5 | **37.9** | 67.1 | **57.6** | **36.8** | **53.6** |
>
> + **Data Scaling vs. VaCo**: We adopt LLaVA (_Vicuna-7B+CLIP_) trained on _LLaVA-665K_ as the baseline, and investigate the effects of more data (_Cambrian-737K_) and VaCo. In the table below, VaCo is more effective than data scaling (_from 665K to 737K_):
>
> | **Method** | **Data Scale** | **MMB_EN** | **MMVP** | **SEED-I** | **Real.** | **MMMU** |
> | --- | :---: | :---: | :---: | :---: | :---: | :---: |
> | LLaVA (baseline) | 665K | 65.5 | 20.0 | 66.0 | 52.7 | 34.5 |
> | LLaVA | 737K | 67.0 | 28.0 | 66.7 | 52.7 | 35.3 |
> | VaCo | 665K | **67.9** | **34.5** | **67.0** | **65.9** | **36.1** |
>
> **Q3: Computational Cost**
>
> **R3**: In the table below, we present the training cost alongside the performance. We conduct _SigLIP+Qwen2-7B_ on _Cambrian-737K_, using 8 H20 GPUs of 96GB memory. Since the vision models are frozen during training, we extract _offline visual features_ for efficient training. VaCo delivers _**significant performance gains without incurring excessive training costs**_:
>
> |  | **Training Cost** | | | | **Performance** | | | | | |
> | --- | :---: | --- | --- | --- | :---: | --- | --- | --- | --- | --- |
> | **Method** | **Trainable Params (B)** | **Speed (s/iter)** | **Time (h)** | **GPU Memory** | **MMB_EN** | **MMVP** | **SEED-I** | **Real.** | **MMMU** | **Average** |
> | LLaVA | 7.611 | 12.63 | ~19 | 55.3 GB | 76.3 | 40.7 | 72.3 | 57.9 | 41.8 | 57.8 |
> | ROSS | 7.696 | 13.56 | ~20 | 60.6 GB | 76.9 | 49.3 | 72.1 | 59.1 | 43.8 | 60.2 |
> | VaCo | 7.875 | 13.65 | ~20 | 61.5 GB | **78.6** | **52.7** | **73.0** | **63.7** | **46.7** | **62.9** |
>
> **Q4: Training Scale and Performance**
>
> **R4**: We investigate the scaling effect of parameters (_from 1.5B to 7B_) and data (_from 665K to 737K_).  Constrained by data and resources, we explored scaling effects preliminarily, yet observed that more parameters and data bring greater improvements:
>
> | **LLM** | **Data Scale**  | **MMB_EN** | **MMVP** | **SEED-I** | **Real.** | **MMMU** | **Average** |
> | --- | :---: | :---: | :---: | :---: | :---: | :---: | :---: |
> | Qwen-1.5B  | 665K  | 70.4 | 30.3 | 68.1 | 56.7 | 40.2 | 53.1 (**+0%**) |
> | Qwen-1.5B | 737K   | 72.2 | 38.3 | 68.9 | 57.3 | 41.3 | 55.7 (**+4.7%**) |
> | Qwen-7B  | 665K   | 77.8 | 47.9 | 72.6 | 61.0 | 46.2 | 61.1 (**+15.1%**) |
> | Qwen-7B | 737K  | **78.6** | **52.7** | **73.0** | **63.7** | **46.7** | **62.9** (**+18.5%**) |
>
> **Q5: Ablation Study for MTQs**
>
> **R5**: With _SigLIP+Qwen2-7B_ on _Cambrian-737K_, we provide the ablation study below:
>
> + **Length of MTQ**. In the initial submission version, we **have already** conducted ablation studies on MTQ length, and the results are provided in Table 10 of the **Appendix**.  For your convenience, we have copied the table from the Appendix:
>
> | **Method** | **MMB_EN** | **MMVP** | **SEED-I** | **Real.** | **MMMU** |
> | --- | :---: | :---: | :---: | :---: | :---: |
> | Length=1 | 66.9 | 35.3 | 66.5 | 55.6 | 35.8 |
> | Length=4 | 67.8 | 36.6 | 66.6 | 56.7 | 35.5 |
> | Length=8 | **68.5** | **37.9** | **67.1** | **57.6** | 36.8 |
> | Length=16 | **68.5** | 36.4 | **67.1** | 57.2 | **37.0** |
>
> Increasing MTQ length from 1 to 8 yields significant performance gains, but further increases bring minimal benefit. As Length=8 is sufficient to capture relevant visual representations, we set it as the default to balance performance and computational cost.
>
> + **Structure of MTQ**. MTQ consists of several trainable tokens placed between visual and text tokens. In the table below, we investigated the MTQ structure.
>
> | **Token Structure** | **MMB_EN** | **MMVP** | **SEED-I** | **Real.** | **MMMU** |
> | --- | :---: | :---: | :---: | :---: | :---: |
> | Vision-Text-MTQ | 67.2 | 33.2 | 65.4 | 54.5 | 34.9 |
> | Vision-MTQ-Text | **68.5** | **37.9** | **67.1** | **57.6** | **36.8** |

---

### Official Review · Reviewer_XRGa · 2025-10-31

**Soundness:** 2
**Presentation:** 3
**Contribution:** 2
**Rating:** 2
**Confidence:** 4

**Summary:**

This paper proposes VaCo, a training method for MLLMs that enhances visual understanding by incorporating supervision from multiple vision foundation models (VFMs). The approach introduces several main components: (1) Modular Task Queries (MTQs) as learnable tokens for specific visual tasks, (2) Visual Alignment Layers (VALs) that project MTQs into task-specific spaces supervised by frozen VFMs, and (3) Token Gateway Mask (TGM) to prevent conflicts between different task queries. The method shows consistent improvements across various benchmarks while maintaining single-encoder inference efficiency.

**Strengths:**

- Motivation is good. Addresses the real limitation that text-only supervision in MLLMs may lead to degradation of visual information,
- This paper gives a novel view that discriminative alignment outperforms generative reconstruction for visual understanding, as shown in Table 4(a).

**Weaknesses:**

- The main concern is that the method is a bit incremental. 1) Learnable queries in MTQ are well-established in [1]. 2) Token gateway mask: TGM is essentially a simple masking strategy without fundamental innovation. The combination feels incremental rather than providing new insights into multimodal learning.
- The perception results in figure 5 are only selected visualization without any quantitative results, which are hard to show the real perceptual quality compared to source VFMs.  This can raise a question that whether MTQs fail to capture task-specific information?

[1] DreamLLM: Synergistic Multimodal Comprehension and Creation. ICLR 2024.

**Questions:**

Please refer to weakness.

---

> ### Author Response · Authors · 2025-11-21
> **Response to Reviewer XRGa**
>
> **Q1: Clarification of Core Insight**
>
> **R1**:  We would like to emphasize that our core insight lies in _**efficiently**_ **integrating multiple visual representations into multimodal understanding**. To address the inference computational drawback of multi-encoder schemes (e.g.,  [2] in Table 8), we employ MTQs to efficiently distill multiple vision models, _facilitating understanding with explicit visual supervision and low inference latency_. Thus, our motivation is _**distinct from learnable queries for generation**_ in unified models:
>
> + _Learnable Query for_ _**Generation**_ _vs. MTQ for_ _**Understanding**_. Although learnable queries for image generation are well-established in unified models (e.g., DreamLLM [1]), they _**serve solely as text-to-image prompts, yielding limited gains for visual understanding**_. In contrast, _**MTQs enhance understanding by integrating multiple visual representations**_ in a lightweight manner. In the Table below, we apply _Vicuna-7B+CLIP_ to demonstrate the performance gap in visual understanding:
>
> | **Method** | **Query Type** | **POPE** | **TextVQA** | **MMBench** | **MM-Vet** |
> | --- | :---: | :---: | :---: | :---: | :---: |
> | LLaVA (baseline) | - | 86.1 | 45.5 | 53.1 | 32.9 |
> | DreamLLM [1] | Dream Query | 83.9 | 41.8 | 58.2 | 36.6 |
> | VaCo (ours) | MTQ | **87.0** | **61.3** | **68.5** | **40.3** |
>
> In Figure 11 of the **Appendix**, we provide attention visualizations to show that MTQs, aligned with multiple visual representations, _achieve better visual understanding_ than the learnable queries in unified models.
>
> + _Masking Strategy for Multi-Objective Alignment_. Besides the efficiency bottleneck, the entangled integration of visual representations [2] hinders improvements in visual understanding for multimodal learning. As shown in Table 4 (b), _**directly stacking multiple MTQs results in feature conflicts among the visual encoders**_. To align stably with multiple visual features, we adopt a masking strategy for disentangling MTQs within the autoregressive framework, facilitating multi-objective alignment for understanding.
>
> **Q2: Quantitative Perceptual Results**
>
> **R2**: In the initial submission version, we **have already** provided the relevant results in Tables 11 and 13 of the **Appendix**. For your convenience, we have copied the tables from the Appendix.
>
> + For 3D evaluation, we use the relative point error (Rel) and the percentage of inliers (δ) on the NYUv2, KITTI, ETH3D, and DIODE:
>
> | **Method** | **NYUv2 Rel ↓** | **NYUv2 δ ↑** | **KITTI Rel↓** | **KITTI δ ↑** | **ETH3D Rel ↓** | **ETH3D δ ↑** | **DIODE Rel ↓** | **DIODE δ ↑** |
> | --- | :---: | :---: | :---: | :---: | :---: | :---: | :---: | :---: |
> | DA V2 | 4.4 | 97.9 | 7.5 | 94.8 | 13.2 | 86.2 | 6.5 | 95.4 |
> | VaCo (**ours**) | 6.9 | 90.2 | 11.8 | 85.1 | 18.7 | 77.6 | 10.3 | 86.0 |
> | VGGT (Depth+Cam) | 3.6 | 98.0 | 9.3 | 91.7 | 4.1 | 97.2 | 27.4 | 78.7 |
> | VaCo (**ours**) | 6.7 | 89.9 | 12.8 | 80.5 | 9.6 | 86.7 | 33.4 | 69.2 |
> | VGGT (Point) | 3.6 | 98.0 | 9.5 | 91.5 | 3.9 | 97.7 | 27.8 | 78.6 |
> | VaCo (**ours**) | 6.6 | 89.9 | 12.9 | 80.1 | 9.3 | 87.3 | 34.3 | 68.1 |
>
> + To evaluate the segmentation results, we report the PQ, AP, and mIoU scores:
>
> | **Method** | **PQ** | **PQthing** | **PQstuff** | **AP** | **mIoU** |
> | --- | :---: | :---: | :---: | :---: | :---: |
> | OneFormer | 57.9 | 64.4 | 48.0 | 49.0 | 67.4 |
> | VaCo (ours) | 43.9 | 48.2 | 33.7 | 37.6 | 53.3 |
>
> [1] Dong, Runpei, et al. "Dreamllm: Synergistic multimodal comprehension and creation." ICLR 2023.
>
> [2] Tong, Peter, et al. "Cambrian-1: A fully open, vision-centric exploration of multimodal llms." NeurIPS 2024.

---

### Author Response · Authors · 2025-12-03
**Summary Comment to AC**

Dear Area Chair,

Thanks for your time and effort in the review process. I am writing to highlight the _core insights_ of our work and summarize our _thorough responses_ to the reviewers' concerns:
___
# Core Insights
Existing MLLMs with only a single CLIP-like encoder often neglect _**diverse perceptual cues (e.g., depth) that are critical for comprehensive visual understanding**_. While several methods integrate multiple visual encoders at the input level, they suffer from (a) progressive _**visual degradation**_ with deeper layers [1] due to a lack of continual visual supervision, and (b) higher _**computational cost**_ (Table 8) caused by the incorporation of multiple encoders. To mitigate visual degradation, we devise a lightweight query-based mechanism to impose _**visual supervision of diverse perceptual cues on MLLM outputs**_,  through efficiently distilling knowledge from multiple visual models. Moreover, for more effective visual supervision, we further _**restrict the interaction among multiple perceptual representations**_ to avoid conflicts. Across various MLLM architectures, our visual integration significantly _**enhances visual understanding performance with negligible computational overhead**_.

In summary, our primary contribution lies in:

+ **We are the first to explore how to enhance visual understanding by efficient output-level supervision from multiple visual encoders, and also firstly coordinate multiple visual representations within MLLM for this purpose**.
___
# Concern Addressing
We have made diligent efforts to address all the concerns raised point by point:
## Reviewer XRGa
- **Q1**: ```Learnable Query for Generation vs. MTQ for Understanding```
- **Q2**: ```Quantitative Perceptual Results```
- **Responses**: (1) We clarify the **differences in motivation and technique** between our MTQ for understanding and learnable queries for generation in unified models [2], and provide experiments showing **our significant understanding gains**; and (2) We showed the **quantitative perceptual results that are already provided in the Appendix**.
___
## Reviewer P8bA
- **Q1**: ```Paper Presentation```
- **Q2**: ```Data/Parameter Scaling vs. VaCo```
- **Q3**: ```Computational Cost```
- **Q4**: ```Training Scale and Performance```
- **Q5**: ```Ablation Study for MTQs```
- **Responses**: (1) We revised the manuscript **in red** by separating the discussion of related work; (2) We provided experiments demonstrating that our method **outperforms simple parameter or data scaling**; (3) Our comparisons show our method achieves **better performance with efficient training and inference**; (4) Additional results showed that our method **exhibits robust scaling effectiveness**; and (5) We showed **MTQ length ablation from the Appendix** and supplement with **MTQ structure ablation**.
___
## Reviewer m47S
- **Q1**: ```Role of Contrastive Loss```
- **Q2**: ```Effectiveness for Recent Architecture```
- **Q3**: ```Perceptual Results and Understanding Performance```
- **Q4**: ```Future improvements```
- **Q5**: ```Input-Level Aggregation vs. Output-Level Supervision```
- **Responses**: (1) We clarified that the **contrastive loss is merely an alignment technology** and re-emphasized our core contributions; (2) We provided additional experiments to show **our effectiveness on SigLIP+Qwen2.5 and SigLIP+Qwen3**; (3) We presented the **positive correlation between perceptual quality and understanding performance**; (4) We discussed future improvements in our response; and (5) We clarified our distinction from the multi-encoder scheme [3] and conducted a comparison to show **our improvements in both understanding and efficiency**.
___
## Reviewer LW6u
- **Q1**: ```Clarification for POPE scores```
- **Q2**: ```Effectiveness for Recent Architecture```
- **Q3**: ```Training Cost```
- **Q4**: ```Loss Selection```
- **Responses**: (1) We reiterated the properties of the POPE metric in our response; (2) Further experimental results confirm **our effectiveness on SigLIP+Qwen2.5 and SigLIP+Qwen3**; (3) We included a comparison of training cost, demonstrating our **significant performance gains with efficient training and inference costs**; and (4) We provided additional quantitative analysis and perceptual results to explain the rationale for loss design.
___
  [1] Wu, Junda, et al. "Mitigating visual knowledge forgetting in MLLM instruction-tuning via modality-decoupled gradient descent." arXiv preprint arXiv:2502.11740.

  [2] Dong, Runpei, et al. "Dreamllm: Synergistic multimodal comprehension and creation." ICLR 2023.

  [3] Tong, Peter, et al. "Cambrian-1: A fully open, vision-centric exploration of multimodal llms." NeurIPS 2024.
___
We believe that these points demonstrate the novelty and significance of our work, and we respectfully request that they be taken into consideration during the evaluation process.

Thanks for your time and attention to this matter.

Best regards,

Authors of Submission2982

---

### Meta-Review · Area_Chair_shjx · 2026-01-06

**Summary:**

This paper presents VaCo, which boosts the models’ visual comprehension capabilities by integrating supervisory signals derived from multiple Vision Foundation Model. The proposed (1) Modular Task Queries (MTQs) serve learnable tokens for specific visual tasks. And Visual Alignment Layers (VALs) project MTQs into task-specific spaces supervised by VFM.

(Reviewer XRGa, Reviewer m47S) The reviewers expresses the comments on the technical novelties (incremental).
(Reviewer P8bA) The paper's writing is hard to follow which needs to be improved.
(Reviewer P8bA and Reviewer ) The training architecture used is out of date, which should use recent published architectures.

I have read the reviewers' comments and have the same feeling on the technical novelties of the paper.
Specifically, the idea of the paper is very similar to the paper namely "DreamVLA: A Vision-Language-Action Model Dreamed with Comprehensive World Knowledge" published in NeurIPS 2025.
The proposed "Modular Task Queries" and "Dream Queries" in Dream VLA are the learnable tokens fed into the LLM.
The proposed "Visual Alignment layer" and "Lightweight Decoder" project LLM embeddings to the visual emebdding space.
As such, the technical novelties of the submitted paper is very limited, which is applied in VLM and VLA.

Considering the concerns of technical novelties, I am considering to reject the paper.

**Reviewer Concerns:**

The readablity of the paper can be addressed by the rebuttal.
The recent publised training architecture can be addressed by the rebuttal.

The technical novelties, which present similar to the published paper DreamVLA, are outstanding, which is also the main concern from myself.

**Reviewer Scores:**

NA

---

### Decision · Program_Chairs · 2026-01-26

Reject